# Hypermetabolism in mice carrying a near-complete human chromosome 21

Dylan C Sarver[1,2], Cheng Xu[1,2], Susana Rodriguez[1,2], Susan Aja[2,3], Andrew E Jaffe[4,5,6,7,8,9], Feng J Gao[1], Michael Delannoy[10], Muthu Periasamy[11,12], Yasuhiro Kazuki[13,14], Mitsuo Oshimura[14], Roger H Reeves[1,8], G William Wong[1,2]*

[1]Department of Physiology, Johns Hopkins University School of Medicine, Baltimore, United States; [2]Center for Metabolism and Obesity Research, Johns Hopkins University School of Medicine, Baltimore, United States; [3]Department of Neuroscience, Johns Hopkins University School of Medicine, Baltimore, United States; [4]Department of Psychiatry and Behavioral Sciences, Johns Hopkins University School of Medicine, Baltimore, United States; [5]Department of Mental Health, Johns Hopkins Bloomberg School of Public Health, Baltimore, United States; [6]The Lieber Institute for Brain Development, Baltimore, United States; [7]Center for Computational Biology, Johns Hopkins University, Baltimore, United States; [8]Department of Genetic Medicine, Johns Hopkins University School of Medicine, Baltimore, United States; [9]Department of Biostatistics, Johns Hopkins Bloomberg School of Public Health, Baltimore, United States; [10]Department of Cell Biology, Johns Hopkins University School of Medicine, Baltimore, United States; [11]Department of Physiology and Cell Biology, The Ohio State University, Columbus, United States; [12]Burnett School of Biomedical Sciences, College of Medicine, University of Central Florida, Orlando, United States; [13]Division of Genome and Cellular Functions, Department of Molecular and Cellular Biology, School of Life Science, Faculty of Medicine, Tottori University, Tottori, Japan; [14]Chromosome Engineering Research Center, Tottori University, Tottori, Japan

*For correspondence: gwwong@jhmi.edu

**Abstract** The consequences of aneuploidy have traditionally been studied in cell and animal models in which the extrachromosomal DNA is from the same species. Here, we explore a fundamental question concerning the impact of aneuploidy on systemic metabolism using a non-mosaic transchromosomic mouse model (TcMAC21) carrying a near-complete human chromosome 21. Independent of diets and housing temperatures, TcMAC21 mice consume more calories, are hyperactive and hypermetabolic, remain consistently lean and profoundly insulin sensitive, and have a higher body temperature. The hypermetabolism and elevated thermogenesis are likely due to a combination of increased activity level and sarcolipin overexpression in the skeletal muscle, resulting in futile sarco(endo)plasmic reticulum $Ca^{2+}$ ATPase (SERCA) activity and energy dissipation. Mitochondrial respiration is also markedly increased in skeletal muscle to meet the high ATP demand created by the futile cycle and hyperactivity. This serendipitous discovery provides proof-of-concept that sarcolipin-mediated thermogenesis via uncoupling of the SERCA pump can be harnessed to promote energy expenditure and metabolic health.

## Editor's evaluation

This important paper provides new insight into the effect of extra-copies of a chromosome, thus aneuploidy, on body metabolisms in mammals. The authors used various solid analyses on the metabolisms and physiology of the transgenic mouse with most of human chromosome 21 and

presented convincing results to support the authors' claims. The work would be of interest to researchers who work on the physiology and biochemistry of body metabolisms in mammals.

## Introduction

The presence of an extra chromosome in mammals is generally lethal during fetal development, due to widespread cellular havoc caused by misregulated gene expression arising from gene dosage imbalance (*Zhu et al., 2018*). Down syndrome (DS), resulting from trisomy of chromosome 21, is one of the rare aneuploidies compatible with life although as many as 80% of trisomy 21 conceptuses miscarry (*Antonarakis et al., 2020*). The increased expression of ~500 transcribed sequences of human chromosome 21 (Hsa21) affects many cell types and organ systems during development and in the postnatal period (*Korenberg et al., 1994*; *Antonarakis, 2017*). Humans with trisomy 21 have cognitive deficits, altered craniofacial development, and are at significantly higher risk for congenital heart defects, hearing and vision loss, leukemia, gastrointestinal disease, and early-onset dementia (*Antonarakis et al., 2020*).

Given the significant impact of intellectual disability on the lives of individuals with DS, research emphasis has naturally focused on the neurological deficits underpinning trisomy 21 (*Potier and Reeves, 2016*). In addition to developmental abnormalities associated with DS, there is an increasing awareness that adolescents and adults with DS also have an increased incidence of obesity, insulin resistance, and diabetes (*Van Goor et al., 1997*; *Bertapelli et al., 2016*; *Fonseca et al., 2005*). Although this was first noted in the 1960s (*Milunsky and Neurath, 1968*), the underlying cause for these metabolic dysregulations is mostly unknown and largely underexplored. Beyond clinical observations, limited studies have been conducted to determine the physiological underpinnings of metabolic impairments seen in DS (reviewed in *Dierssen et al., 2020*). Our recent study on the Ts65Dn mouse model represents the most in-depth metabolic analysis, to date, of any DS mouse model (*Sarver et al., 2023*). However, the segmental trisomic Ts65Dn mouse contains only ~55% of the orthologous protein-coding genes (PCGs) found on Hsa21 (*Gupta et al., 2016*). In addition, it contains additional trisomic genes from the centromeric region of mouse chromosome 17 (Mmu17) not found in Hsa21, thus complicating the genotype-phenotype relationships (*Duchon et al., 2011*; *Reinholdt et al., 2011*).

In the past two decades, more than 20 mouse models of DS have been generated (*Herault et al., 2017*). Despite their utility in advancing DS research, none of these models recapitulate the full spectrum of human DS. With the exception of Tc1, all the DS mouse models are trisomic for some, but not all, of the orthologous mouse genes found in Hsa21 (*Herault et al., 2017*). Tc1 is the first mouse model with an independently segregating Hsa21 (*O'Doherty et al., 2005*). However, Tc1 mice are missing >50 of the 220 PCGs on Hsa21 due to deletion and mutations (*Gribble et al., 2013*). In addition, Tc1 mice show extensive mosaicism (i.e. the human chromosome is present in zygotes but is lost randomly from cells during development). As a consequence, every mouse has a unique developmental trajectory, complicating the interpretations of results obtained from Tc1 mice. To overcome the limitations of previously generated trisomic mouse models, a transchromosomic mouse model (TcMAC21) carrying an independently segregating and near-complete copy of Hsa21 was recently developed (*Kazuki et al., 2020*). TcMAC21 is not mosaic and contains 93% of HSA21q PCGs, and is considered the most representative mouse model of DS.

Both mouse and rat that carry a non-mosaic Hsa21 recapitulate many DS phenotypes related to the central nervous system (e.g. reduced cerebellum volume, learning and memory deficit), craniofacial skeleton, and heart (*Kazuki et al., 2020*; *Kazuki et al., 2022*). The metabolic phenotype of TcMAC21, however, is unknown and has yet to be examined. The availability of the TcMAC21 mouse model has afforded a unique opportunity to address two fundamental questions: (1) what is the impact of aneuploidy on systemic metabolism; (2) what are the molecular, cellular, and physiological consequences of introducing a foreign (human) chromosome from an evolutionarily distant species into mice?

Unexpectedly, we discovered that TcMAC21 mice have all the hallmarks of hypermetabolism, likely driven by a combination of hyperactivity and elevated mitochondrial respiration and futile sarco(endo)plasmic reticulum $Ca^{2+}$ ATPase (SERCA) pump activity in the skeletal muscle as a consequence of endogenous sarcolipin (SLN) overexpression. Our study has provided further evidence and

proof-of-concept that endogenous SLN-mediated uncoupling of the SERCA pump can be harnessed for energy dissipation, weight loss, and metabolic health.

## Results

### Human chromosome 21 genes are differentially expressed and regulated in mouse adipose tissue, liver, and skeletal muscle

TcMAC21 mice carry a non-mosaic and independently segregating mouse artificial chromosome with a near-complete copy of the long arm of human chromosome 21 (Hsa21q) (*Kazuki et al., 2020*). The Hsa21q in TcMAC21 is comprised of ~37 Mb and 199 PCGs (*Figure 1A*). RNA-sequencing showed that TcMAC21 mice are capable of expressing Hsa21-derived transcripts in each of the tissues examined, and that gene expression is regulated in a tissue-specific manner. The transcriptional activity map of Hsa21 shows regions of gene expression and repression (*Figure 1B*). There are three large regions of Hsa21 with little or no transcription activity: 29.5–31.1 Mb, 44.5–44.8 Mb, and 45.3–46.2 Mb. The first gap of transcriptional inactivity contains the PCGs *Cldn8* and *Cldn17*, *Girk1*, and 33 distinct *Krtap* (keratin-associated protein) genes. The second transcriptionally inactive area contains 16 *Krtap* genes, and the third transcriptionally inactive area contains 8 PCGs (*Col6a1, Col6a2, Col18a1, Fctd, Lss, Pcbp3, Slc19a1,* and *Spatc1l*). By filtering the transcriptional map to display expressed genes only, we highlighted all the genes with their differential expression profiles across five major metabolic tissues—brown adipose tissue (BAT), inguinal white adipose tissue (iWAT), gonadal white adipose tissue (gWAT), liver, and skeletal muscle (*Figure 1C*).

One of the more striking differences in expression profile is between visceral (gonadal) and subcutaneous (inguinal) white adipose tissue (gWAT and iWAT respectively). gWAT expresses 115 human PCGs while iWAT expresses only 27. A similar pattern can be seen in the non-protein-coding genes (NPCGs). We were unable to detect any Hsa21-derived NPCGs in the iWAT, while in gWAT we observed 37. Overlap analysis was carried out to assess how similar expression profiles were between tissues (*Figure 1D*). Of the 126 Hsa21-derived PCGs expressed by at least one tissue, a majority (65 total) are shared between BAT, gWAT, liver, and skeletal muscle. Of the 109 Hsa21-derived NPCGs expressed by at least one tissue, the majority (53 total) are uniquely expressed by skeletal muscle. Of note, the liver uniquely expresses 6 PCGs and 2 NPCGs, gWAT 4 and 3, skeletal muscle 3 and 53, BAT 1 and 10, and iWAT 0 and 0. Together, these data indicate that Hsa21-derived transcripts are differentially expressed and regulated across major metabolic tissues in TcMAC21 mice.

### Hypermetabolism in TcMAC21 mice fed a standard chow

Having established that Hsa21-derived transcripts are differentially expressed and regulated in mouse organs and tissues, we next asked the impact of the extra human genetic material and genes on systemic metabolism. As previously documented, TcMAC21 pups are born at the same weight as their euploid littermates (*Kazuki et al., 2020*). However, by 3 months of age TcMAC21 mice fed a standard chow weighed significantly less (~8.5 g) than euploid littermates, and this weight difference remained stable over time (*Figure 2A*). The size and body weight differences were not due to reduced plasma IGF-1 and growth hormone, as their circulating levels were in fact higher in TcMAC21 compared to euploid mice (*Figure 2—figure supplement 1*). Body composition analysis showed that TcMAC21 have significantly reduced absolute and relative (normalized to body weight) fat mass (*Figure 2B*). Although the absolute lean mass was reduced in TcMAC21 mice, the relative lean mass (normalized to body weight) was not different between genotypes (*Figure 2B*). Tissue collection at termination of the study also showed smaller visceral and subcutaneous fat mass and liver weight in TcMAC21 mice (*Supplementary file 1*).

Differences in body weight were not due to reduced caloric intake, as TcMAC21 mice actually consumed the same or slightly higher amount of food despite being markedly leaner (*Figure 2C* and *Supplementary file 2*). Thus, relative to their body weight, TcMAC21 actually consumed a significantly higher amount of food than euploid controls (*Figure 2C*). Indirect calorimetry analysis indicated that TcMAC21 mice—regardless of the photocycle (light or dark phase) and metabolic states (ad libitum fed, fasting, refeed)—were expending ~25% more energy and were significantly more active compared to euploid controls (*Figure 2D–F* and *Supplementary file 2*). It is known that normalization to lean mass can lead to an overestimation of energy expenditure (EE) (*Tschöp et al., 2012*). For this

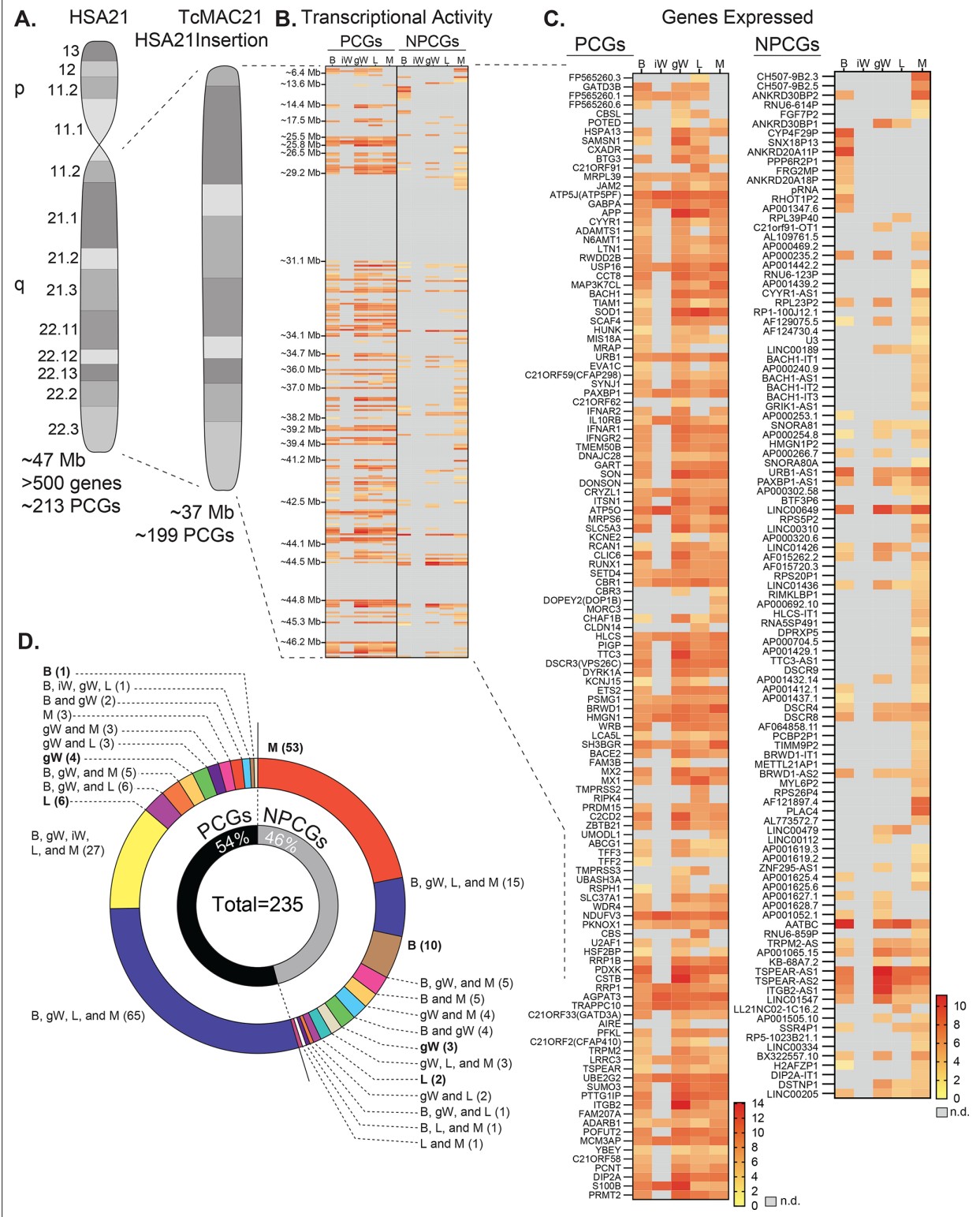

**Figure 1.** Human chromosome 21 genes are differentially expressed and regulated in mouse adipose tissue, liver, and skeletal muscle. (**A**) Graphical representation of human chromosome 21 (Hsa21) and the entire long arm (Hsa21q) region carried by a mouse artificial chromosome in the transchromosomic mouse model (TcMAC21). Four deletions that occurred during generation of the transchromosomic mice eliminate 14/213 protein-coding genes (PCGs; 7%) and 105/487 predicted or known non-protein-coding genes (NPCGs; 22%) (*Kazuki et al., 2020*). (**B**) Global view of transcriptionally expressed and repressed PCG and NPCG regions over the entire Hsa21q across five tissues. Gray box denotes transcript that is not detected. (**C**) Transcriptional activity map showing only Hsa21 genes expressed by at least one tissue. Gray box denotes transcript that is not detected.

*Figure 1 continued on next page*

*Figure 1 continued*

(**D**) Overlap analysis showing shared expression of human PCGs and NPCGs across five tissues. Of the 235 unique human genes expressed by the TcMAC21 mice, 54% are PCGs and 46% are NPCGs. B, brown adipose tissue; iW, inguinal white adipose tissue; gW, gonadal white adipose tissue; L, liver; M, skeletal muscle (gastrocnemius); n.d., not detected. n=5 RNA samples per group per tissue type. Mice were on high-fat diet for 16 weeks at the time of tissue collection.

reason, ANCOVA (using body weight as a covariate of EE) (*Tschöp et al., 2012*) were also performed. Both types of analyses suggested that TcMAC21 mice had significantly higher EE relative to euploid controls (*Figure 2D–F*). Despite much higher caloric intake per gram body mass, TcMAC21 mice were much leaner due to substantially elevated physical activity and EE. Hyperactivity and elevated EE were not due to altered circulating thyroid hormones, as serum triiodothyronine (T$_3$, the active form of TH) levels were not different between chow fed TcMAC21 and euploid mice (*Figure 2—figure supplement 1*). Serum level of thyroxine (T$_4$), the precursor of T$_3$, were modestly elevated in TcMAC21 relative to euploid mice.

In accordance with the lean phenotype, TcMAC21 mice had significantly smaller adipocyte cell size in both subcutaneous (inguinal) and visceral (gonadal) fat depots (*Figure 2G–H*), as well as significantly reduced fat accumulation in liver (*Figure 2I*). Fasting triglyceride, non-esterified fatty acid (NEFA), and β-hydroxybutyrate levels were not different between genotypes; fasting cholesterol, however, was higher in TcMAC21 mice (*Figure 2J*). Although fasting insulin levels were not different between groups, fasting blood glucose was significantly lower in TcMAC21 mice (*Figure 2K*). The insulin resistance index (homeostatic model assessment for insulin resistance [HOMA-IR]) along with GTT and insulin tolerance test (ITT) suggested modest improvements in insulin sensitivity in TcMAC21 relative to euploid mice (*Figure 2L–N*). Assessment of the pancreas showed that TcMAC21 mice have similar β-islet cross-sectional area (CSA), insulin and somatostatin (SST) content, and insulin granule and vesicle size compared to euploid controls (*Figure 2—figure supplement 2*). Taken together, these data indicate that chow-fed TcMAC21 mice at baseline are lean despite increased caloric intake, and this is largely due to elevated physical activity and EE.

## TcMAC21 mice are resistant to diet-induced obesity and metabolic dysfunction

The hypermetabolic phenotypes seen in chow-fed TcMAC21 predicted that these mice would be resistant to diet-induced obesity and metabolic dysfunction. Indeed, after 8 weeks on high-fat diet (HFD), TcMAC21 mice gained only ~3 g of body weight, whereas the euploid controls gained >15 g of body weight over the same period. Consequently, TcMAC21 mice weighed ~50% less than euploid controls (*Figure 3A*). Consistent with the lean phenotype, the absolute and relative (normalized to body weight) fat mass were markedly reduced compared to euploid controls (*Figure 3B*). The weights of other organs (liver, kidney, BAT) at time of termination were also lower in TcMAC21 mice, but tibia length was not different between genotypes (*Figure 3—figure supplement 1* and *Supplementary file 3*). Complete blood count revealed no differences in erythroid, lymphoid, and myeloid cell numbers between genotypes (*Supplementary file 4*). Because relative lean mass was higher in TcMAC21 compared to euploid mice (*Figure 3B*), the lean phenotype seen in HFD-fed TcMAC21 is largely due to reduced adiposity. Accordingly, TcMAC21 had significantly smaller adipocyte cell size in both subcutaneous (inguinal) and visceral (gonadal) fat depots, and a marked reduction in lipid accumulation in the liver (*Figure 3C–E*).

Although fasting serum triglyceride, NEFA, and β-hydroxybutyrate levels were not different between genotypes, serum cholesterol was significantly lower in TcMAC21 mice (*Figure 3F*). Fasting glucose and insulin levels, and the insulin resistance index (HOMA-IR), were markedly lower in TcMAC21 mice relative to euploid controls (*Figure 3G–H*), indicative of enhanced insulin sensitivity. In glucose tolerance tests (GTTs), even though the rate of glucose disposal was similar between TcMAC21 and euploid mice, the amount of serum insulin present during GTT (time 0, 15, and 30 min) was dramatically lower in TcMAC21 (*Figure 3I–J*). This indicates that a substantially lower amount of insulin is sufficient to promote glucose clearance in TcMAC21 at a rate comparable to euploid mice, consistent with elevated insulin sensitivity in the peripheral tissues. Indeed, when we directly assessed insulin action via ITTs, TcMAC21 mice clearly exhibited higher insulin sensitivity as indicated by the significant differences in insulin-stimulated glucose disposal (*Figure 3K*).

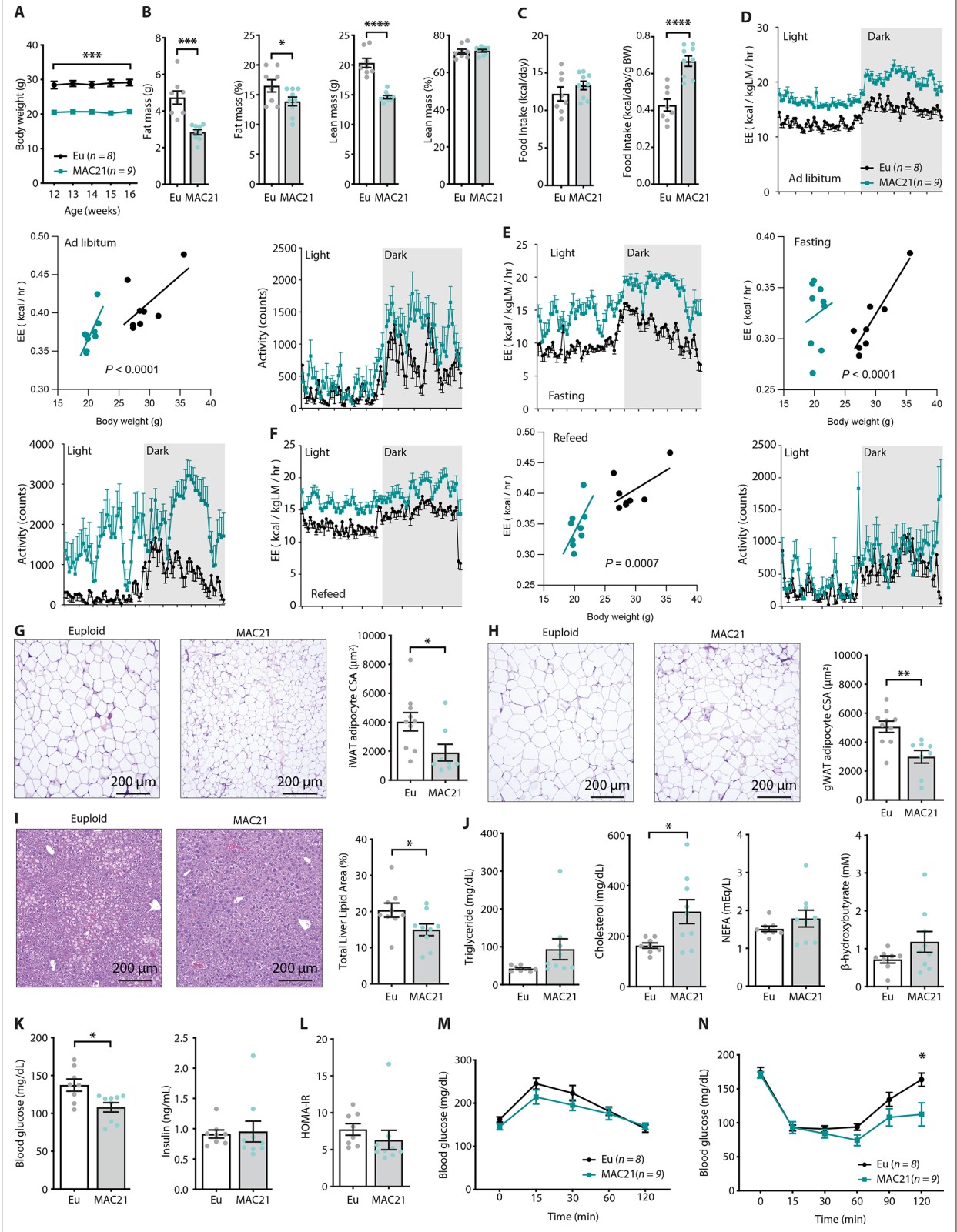

**Figure 2.** Hypermetabolism in TcMAC21 mice fed a standard chow. (**A**) Body weights of mice fed a standard chow. (**B**) Body composition analysis of fat and lean mass (relative to body weight). (**C**) Absolute and relative (normalized to body weight) food intake over a 24 hr period. (**D–F**) Energy expenditure (EE) and physical activity level over 24 hr period in ad libitum chow-fed mice (**D**), during fasting (**E**), and refeeding after a fast (**F**). EE is normalized to lean mass in the 24 hr trace or analyzed by ANCOVA where body weight was used as a covariate. (**G**) Hematoxylin and eosin (H&E)-stained sections

*Figure 2 continued on next page*

Figure 2 continued

of inguinal white adipose tissue (iWAT) and adipocyte cross-sectional area (CSA) quantification. (**H**) Histology of gonadal white adipose tissue (gWAT) and adipocyte CSA quantification. (**I**) Histology of liver tissues with quantification of area covered by lipid droplets per focal plane. (**J**) Fasting serum triglyceride, cholesterol, non-esterified fatty acids (NEFA), β-hydroxybutyrate (ketone) levels. (**K**) Fasting blood glucose and insulin levels. (**L**) Insulin resistance index (homeostatic model assessment for insulin resistance [HOMA-IR]). (**M**) Glucose tolerance tests. (**N**) Insulin tolerance tests. Sample size for all data: euploid (n=8) and TcMAC21 (n=9).

The online version of this article includes the following figure supplement(s) for figure 2:

**Figure supplement 1.** ELISA data from euploid and MAC21 mice.

**Figure supplement 2.** Electron micrograph quantification of pancreatic islet and acinar cell zymogen granules.

**Figure supplement 3.** Light and dark cycle body temperature of euploid and MAC21 mice fed a standard chow.

To independently confirm TcMAC21 mice are more insulin sensitive, we fasted the mice overnight (16 hr) then reintroduced them to food. Under this fasting-refeeding condition, we could clearly see the resumption of food intake was successful at increasing blood glucose in TcMAC21 (*Figure 3L*); however, the insulin response to food intake in TcMAC21 mice was strikingly smaller in magnitude compared to euploid controls (*Figure 3M*). Again, these data indicate that TcMAC21 mice are significantly more insulin sensitive since a substantially lower insulin response during fasting-refeeding is sufficient for glucose clearance at a rate comparable to euploid mice.

These results prompted us to determine if there were developmental changes in the pancreas leading to reduced insulin secretion in response to glucose administration or food intake, independent of elevated insulin sensitivity in peripheral tissues. Quantification of β-islet size, pancreatic insulin and SST content, as well as insulin granule and vesicle size did not reveal any intrinsic differences between TcMAC21 and euploid mice (*Figure 3N–Q*), thus ruling out a developmental cause and arguing in favor of enhanced insulin action. Pancreatic acinar zymogen granule size was also not different between genotypes (*Figure 2—figure supplement 2*), suggesting normal development of the exocrine pancreas. Taken together, these data indicate that TcMAC21 mice are remarkably resistant to HFD-induced obesity and insulin resistance.

## Hypermetabolism in TcMAC21 mice fed an HFD

Next, we sought to uncover the physiological mechanisms responsible for TcMAC21 resistance to weight gain and developing insulin resistance when fed an HFD. First, we wanted to rule out whether there is a change in caloric intake. TcMAC21 mice actually consumed the same or slightly higher amount of food as the euploid controls despite markedly lower body weight (*Figure 4A* and *Supplementary file 5*). Thus, relative to their body weight, HFD-fed TcMAC21 mice consumed a significantly higher amount of food than the euploid controls (*Figure 4A*). To rule out any potential dysfunction of the gut that might adversely affect nutrient absorption, we collected, counted, weighed, and subjected fecal samples of each mouse to fecal bomb calorimetry. Neither fecal frequency, average fecal pellet weight, nor fecal energy content was different between TcMAC21 and euploid mice (*Figure 4B*). Fecal energy content trended lower in TcMAC21 (*Figure 4B*), implying a modest increase in the efficiency of nutrient absorption. Since energy input (caloric intake per gram body weight) was significantly higher in TcMAC21 and outputs (left-over fecal energy) were similar across genotypes, the data suggest hypermetabolism being a cause of the lean phenotype seen in TcMAC21 mice. Indeed, when we measured EE and physical activity of both groups, we found that TcMAC21 mice have markedly higher EE and physical activity irrespective of circadian cycle and metabolic states (*Figure 4D–F* and *Supplementary file 5*). Aware of the potential overestimation of EE when normalized to lean mass, we also included ANCOVA where body weight was used a covariate (*Tschöp et al., 2012*). Both types of analyses suggested that TcMAC21 mice fed an HFD had significantly higher EE relative to euploid controls (*Figure 4D–F*). The striking difference in EE was very similar to TcMAC21 fed a standard chow (*Figure 2D–F*), but to an even greater extent when mice were fed an HFD, presumably due to the greater availability of calorie-dense lipid substrates for oxidation.

Because TcMAC21 mice burned a large excess of energy, we measured the circulating levels of thyroid hormones as they are known to increase metabolic rate and EE (*Mullur et al., 2014*). Both serum $T_3$ (the active form) and $T_4$ (precursor of $T_3$) levels were significantly higher in TcMAC21 relative to euploid mice (*Figure 4C*). $T_3$ hormone, however, was not elevated in HFD-fed mice housed at

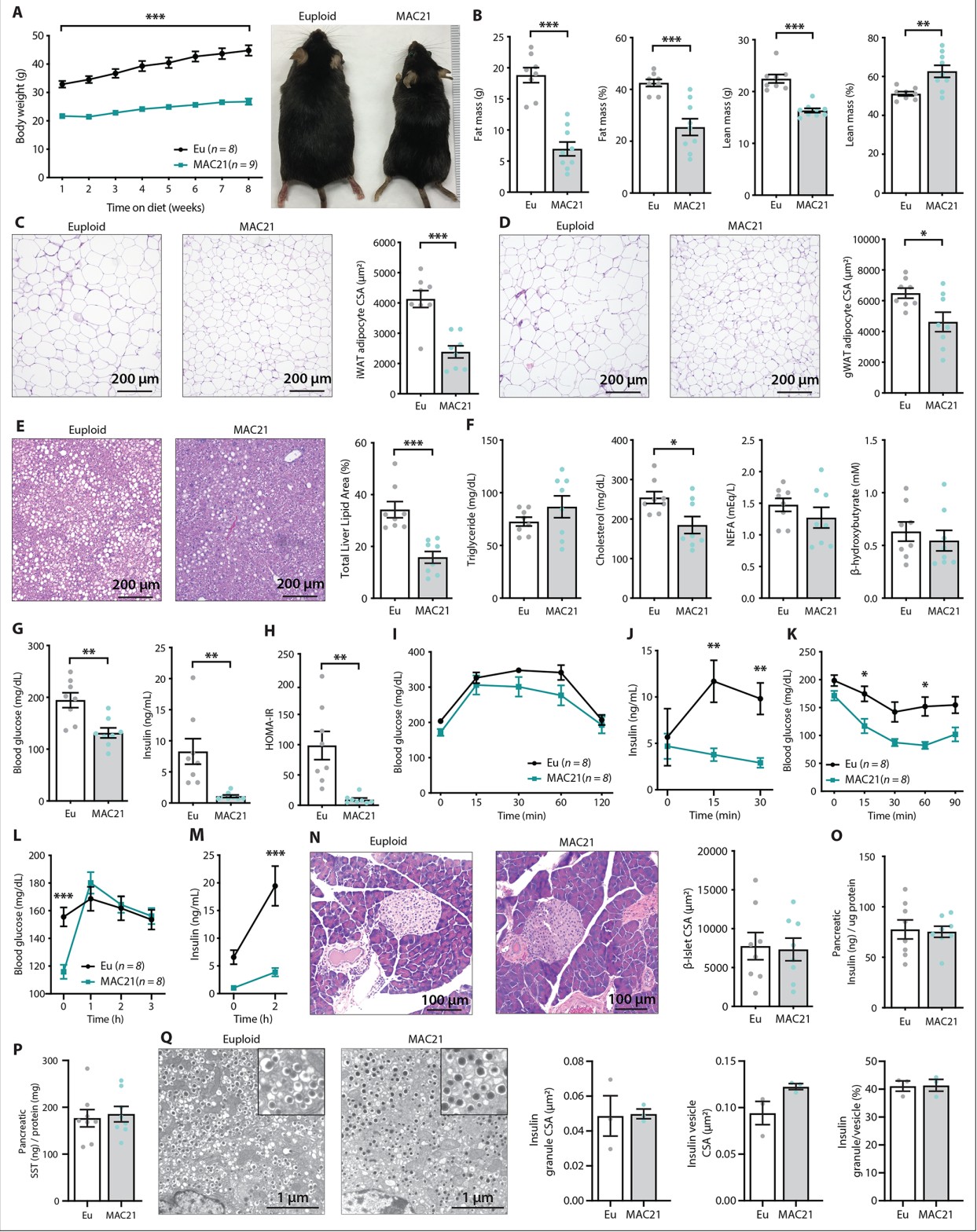

**Figure 3.** TcMAC21 mice are resistant to diet-induced obesity and metabolic dysfunction. (**A**) Body weights over time on a high-fat diet and representative mouse images. (**B**) Body composition analysis of fat and lean mass. (**C**) Histology of inguinal white adipose tissue (iWAT) and quantification of adipocyte cross-sectional area (CSA). (**D**) Histology of gonadal white adipose tissue (gWAT) and quantification of adipocyte CSA. (**E**) Histology of liver tissue and quantification of area covered by lipid droplets per focal plane. (**F**) Fasting serum triglyceride, cholesterol, non-esterified fatty acids (NEFA), β-hydroxybutyrate (ketone) levels. (**G**) Fasting blood glucose and insulin levels. (**H**) Insulin resistance index (homeostatic model

*Figure 3 continued on next page*

*Figure 3 continued*

assessment for insulin resistance [HOMA-IR]). (**I**) Glucose tolerance tests (GTTs). (**J**) Serum insulin levels during GTT. (**K**) Insulin tolerance tests (ITTs). (**L**) Blood glucose levels after an overnight (16 hr) fast and 1, 2, and 3 hr of food reintroduction. (**M**) Serum insulin levels after a 16 hr fast and 2 hr of refeeding. (**N**) Pancreas histology and quantification of β-islet CSA. (**O–P**) Pancreatic insulin and somatostatin (SST) contents (normalized to pancreatic protein input). (**Q**) Electron micrographs (EM) of pancreatic β-cells showing dense insulin granules and their surrounding vesicles, and quantification of insulin granule CSA, insulin vesicle CSA, and the ratio of insulin granule to insulin vesicle. n=8 euploid and 8–9 TcMAC21 mice for all graphs from A to P. n=3 euploid and 3 TcMAC21 used for EM quantification; each data point represents 1200 insulin granules and 1200 insulin vesicles quantified across six unique locations per mouse, graphs Q.

The online version of this article includes the following figure supplement(s) for figure 3:

**Figure supplement 1.** Representative tissue dissection images of euploid and MAC21 mice fed a high-fat diet.

thermoneutrality (30°C) (*Figure 2—figure supplement 1*). If EE was elevated, body temperature of TcMAC21 mice would likely increase. Indeed, deep colon temperatures of TcMAC21 were elevated, most notably in the dark cycle when mice are active (*Figure 4G*). Assessment with thermal imaging showed an elevated skin temperature around the interscapular region of TcMAC21 mice, whereas the tail skin temperature was not different between groups (*Figure 4G–H*). Importantly, the differences in interscapular skin temperatures persisted in TcMAC21 even when compared to weight-matched wild-type mice (*Figure 4I–J*), thus ruling out body weight (and hence surface area/volume ratio) as the cause of greater heat generation to compensate for greater heat loss. Elevated body temperature, however, was not observed in chow-fed mice (*Figure 2—figure supplement 3*), even though chow-fed TcMAC21 were also hyperactive and had higher EE. Consistent with the thermal imaging data, histological analysis of the interscapular BAT revealed a marked reduction in fat accumulation and a 'healthy' brown appearance in TcMAC21 mice when compared to euploid controls (*Figure 4K–L*), presumably due to excess lipids being utilized. We therefore expected several key metabolic genes in BAT to be upregulated in TcMAC21. Surprisingly, we found minimal differences in key thermogenic and fat oxidation genes between the two groups of mice (*Figure 4M*, *Figure 5—figure supplement 1*).

## Hypermetabolism of TcMAC21 mice is uncoupled from changes in adipose and liver transcriptomes

The lack of changes observed in key thermogenic and fat oxidation genes mentioned above led us to assume the differences in gene expression must be due to non-canonical and potentially novel pathways. To test this, we conducted an unbiased RNA-sequencing analysis of BAT, liver, gWAT, and iWAT. Again, to our surprise and contrary to expectation, we found the transcriptomes of BAT, liver, gWAT, and iWAT in TcMAC21 to be remarkably similar to euploid controls, with only a limited number of genes whose expression was significantly altered (*Figure 5A–D*).

There was not a single tissue that had more than 0.3% of its transcriptome (mouse) significantly altered (*Figure 5E* and *Supplementary files 6-9*). None of the significant changes in gene expression—small in number—could readily account for the striking differences in phenotypes between TcMAC21 and euploid mice. Corroborating the RNA-sequencing results, quantitative real-time PCR analyses of select metabolic genes in BAT, liver, gWAT, and iWAT also showed minimal changes in TcMAC21 regardless of diet (chow or HFD) and temperature (22°C or 30°C) (*Figure 5—figure supplements 1–5*). Together, these data indicate that TcMAC21 mice are hyperactive and hypermetabolic with elevated thermogenesis, but these phenotypes are largely uncoupled from transcriptomic changes in BAT, liver, gWAT, and iWAT.

## SLN overexpression in skeletal muscle drives TcMAC21 hypermetabolism

Given the remarkable differences seen in body weight, adiposity, tissue histology and lipid contents, insulin sensitivity, body temperature, physical activity, and metabolic rate between TcMAC21 and euploid mice, we were surprised by how few changes in transcriptomes—both in number and magnitude—were occurring in BAT, liver, and two major white adipose depots. This prompted us to examine the transcriptome of skeletal muscle, by far the largest metabolically active tissue that can substantially contribute to overall EE. RNA-sequencing of TcMAC21 skeletal muscle (gastrocnemius) revealed

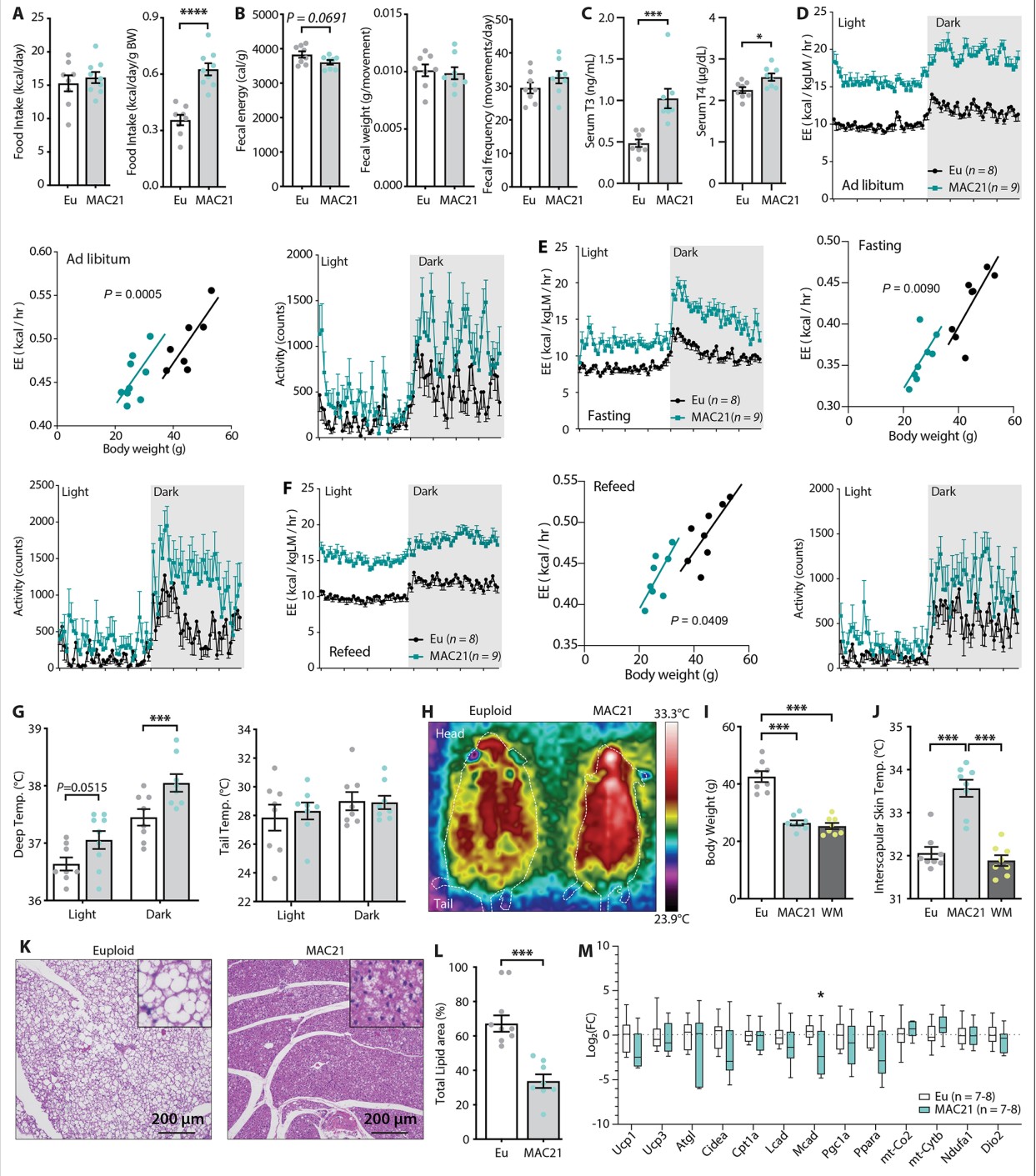

**Figure 4.** Hypermetabolism in TcMAC21 mice fed a high-fat diet (HFD). (**A**) Absolute and relative (normalized to body weight) food intake in mice fed an HFD. (**B**) Fecal energy content, fecal weight, and fecal frequency of TcMAC21 mice and euploid controls. (**C**) Serum triiodothyronine ($T_3$) and thyroxine ($T_4$) levels. (**D–F**) Energy expenditure (EE) and activity level over 24 hr period in ad libitum HFD-fed mice (**D**), during fasting (**E**), and refeeding after a fast (**F**). EE is normalized to lean mass in the 24 hr trace or analyzed by ANCOVA where body weight was used as a covariate. (**G**) Deep colonic and tail temperature measured over 3 days in both the light and dark cycle. (**H**) Representative infrared images of mice. (**I**) Body weights of euploid, TcMAC21, and weight-matched (WM) control C57BL/6 mice. (**J**) Interscapular skin temperature of euploid, TcMAC21, and WM control mice. (**K**) Representative histology of brown adipose tissue (BAT). (**L**) Quantification of percent total lipid area coverage per focal plane in BAT of euploid and TcMAC21. (**M**) Expression of mouse genes (by qPCR) known to play major metabolic roles in BAT. Sample size for all data: euploid (n=8) and TcMAC21 (n=9).

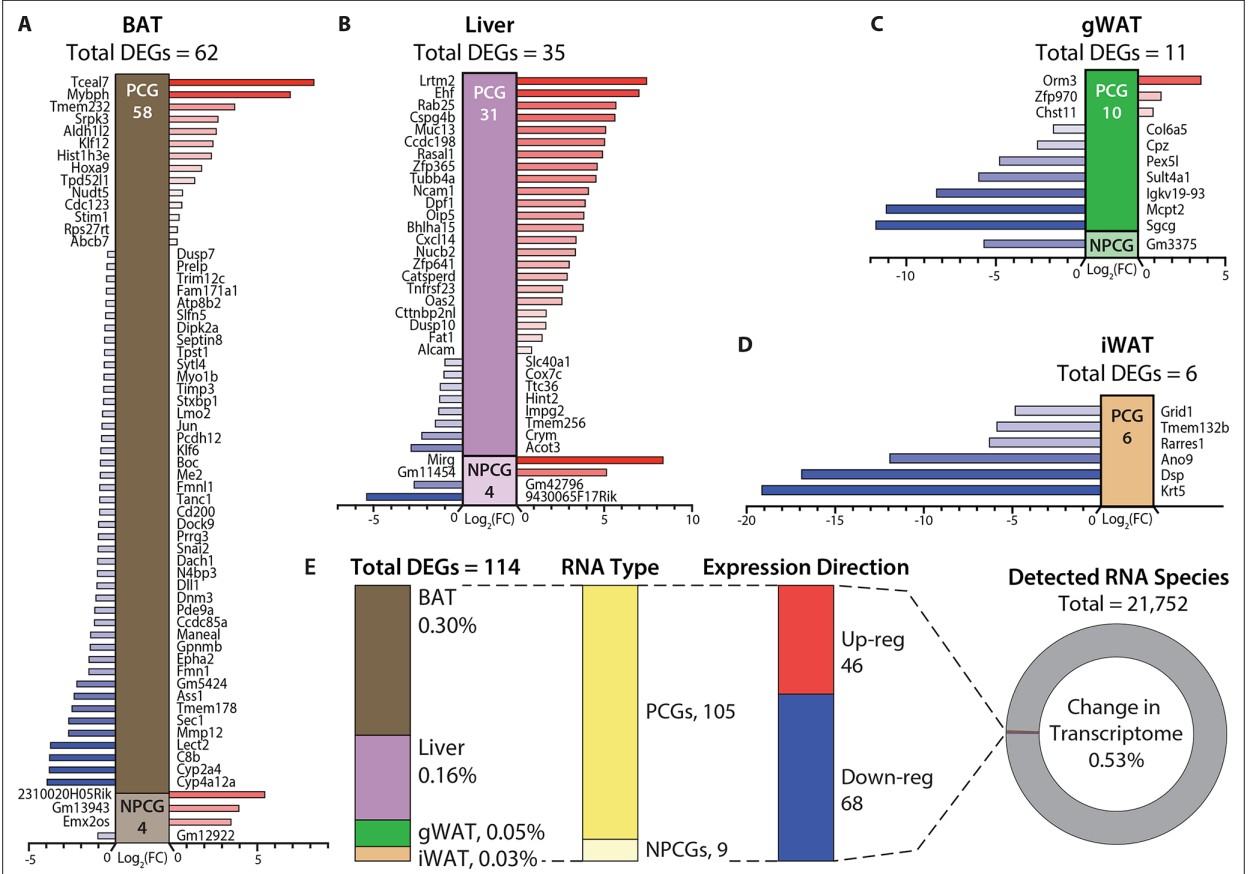

**Figure 5.** Hypermetabolism of TcMAC21 mice is uncoupled from changes in adipose and liver transcriptomes. (**A–D**) Differentially expressed mouse genes (DEGs), both protein-coding (PCG) and non-protein-coding (NPCG) in brown adipose tissue (BAT), liver, gonadal white adipose tissue (gWAT), and inguinal white adipose tissue (iWAT). All data is relative to euploid and presented as Log2(FC). The list of genes represents all up- and down-regulated mouse genes (significant by adjusted p-value cut-off) for all four tissues. The red bars indicate upregulated genes and the blue bars indicate downregulated genes. (**E**) General view and summary of transcriptional changes in BAT, liver, gWAT, and iWAT to highlight the strikingly minimal changes in the mouse transcriptome across the four tissues. There are only a combined total of 114 differentially expressed genes (DEGs) across four tissues, with the relative percentage (out of the 114 DEGs) shown for each tissue. Of the 114 DEGs, 105 are protein-coding genes (PCGs; dark yellow bar) and 9 are non-protein-coding genes (NPCGs; light yellow bar). Of the 114 DEGs, 46 are upregulated (red bar) and 68 are downregulated (blue bar). In total, only a combined 0.53% change is noted in the transcriptome of all four tissues (out of the 21,752 RNAs detected). Sample size for RNA-sequencing: euploid (n=5 per tissue) and TcMAC21 (n=5 per tissue).

The online version of this article includes the following figure supplement(s) for figure 5:

**Figure supplement 1.** Volcano plot of all qPCR results from brown adipose tissue (BAT), skeletal muscle, and white adipose tissue.

**Figure supplement 2.** Gene expression analysis in brown adipose tissue.

**Figure supplement 3.** Gene expression analysis in inguinal white adipose tissue (iWAT).

**Figure supplement 4.** Gene expression analysis in gonadal white adipose tissue (gWAT).

**Figure supplement 5.** Gene expression analysis in liver.

it to be the most transcriptionally dynamic tissue, with 4.2% of the transcriptome changed relative to euploid controls (*Figure 6A*). Within the skeletal muscle of TcMAC21, there were 432 upregulated genes and 501 downregulated genes (*Figure 6—figure supplement 1* and *Supplementary file 10*). Gene Ontology analysis of the differentially expressed genes (DEGs) highlighted upregulated pathways related to thermogenesis, mitochondrial activity, and amino acid metabolism; and downregulated pathways related to TGF-β, insulin, growth hormone, calcium, adrenergic, serine/threonine kinase, and cGMP-PKG signaling pathways (*Figure 6B*).

To further confirm the RNA-sequencing results, we performed qPCR on a number of key genes related to skeletal muscle metabolism, fast- and slow-twitch fiber types, futile cycling, thyroid hormone

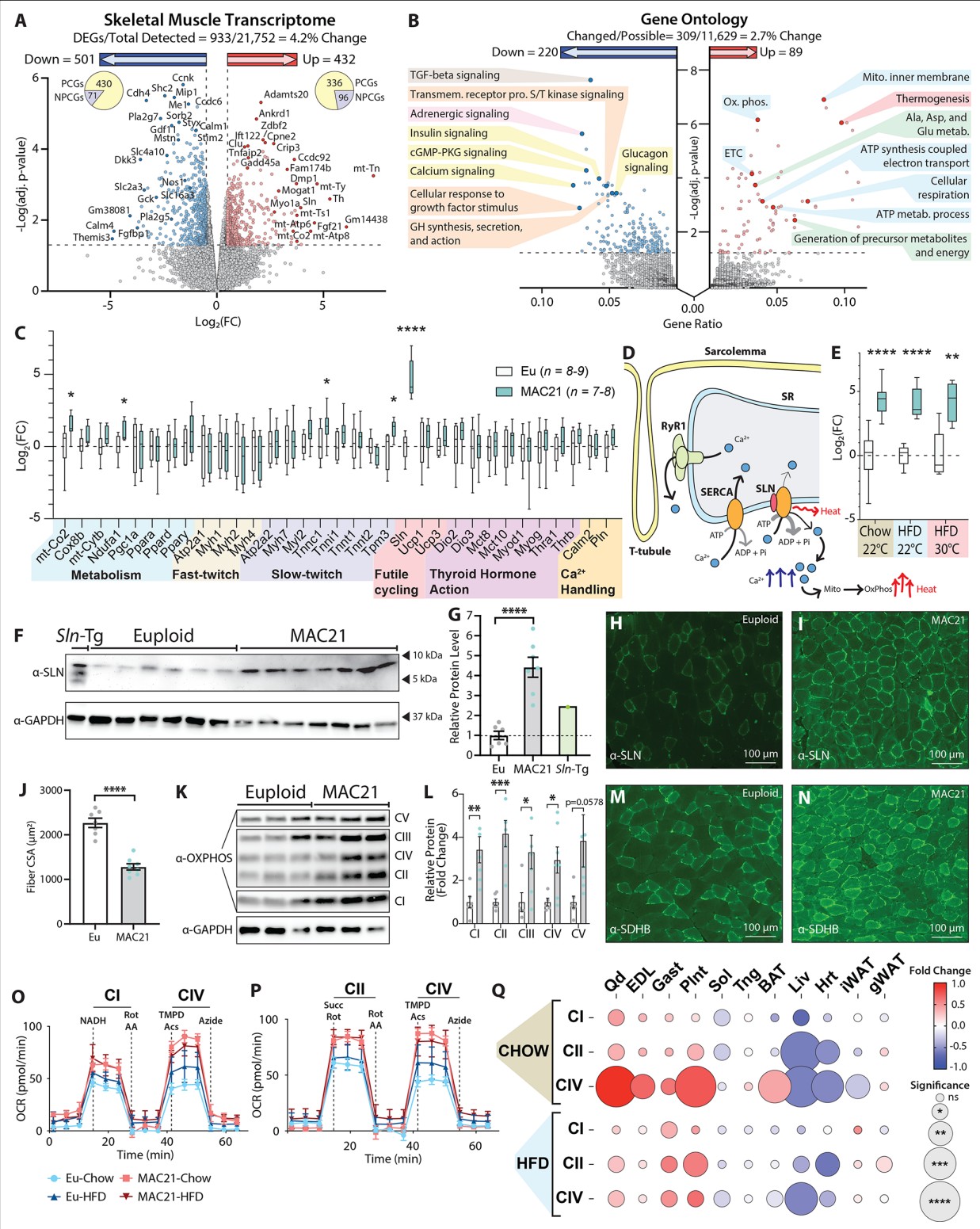

**Figure 6.** Sarcolipin overexpression in skeletal muscle drives TcMAC21 hypermetabolism. (**A**) RNA-sequencing analysis reveals changes in TcMAC21 relative to euploid skeletal muscle (gastrocnemius). Volcano plot of skeletal muscle transcriptome (transcripts from the mouse genome only). The lower dotted line denotes significance at the adjusted p-value cut-off (adj. p=0.05). The vertical dotted lines denote a Log2(FC) of –0.5 or 0.5. (**B**) Gene Ontology analysis of RNA-sequencing results using ClusterProfiler (*Yu et al., 2012*). (**C**) Expression of genes (by qPCR) known to be involved in metabolism, fast- or slow-twitch fiber types, futile cycling, thyroid hormone action, and calcium handling in skeletal muscle (gastrocnemius). (**D**) Graphical representation of Ca²⁺ cycling through the sarcoplasmic reticulum (SR). Not shown is the store-operated calcium entry pathway involving

*Figure 6 continued on next page*

*Figure 6 continued*

Ca$^{2+}$ channels located at the plasma membrane/T-tubule. Ca$^{2+}$ leaks out of the SR through the ryanodine receptor (RyR1) and is re-sequestered back to the SR by the sarco(endo)plasmic reticulum Ca$^{2+}$ ATPase (SERCA). SERCA pump uses the energy derived from ATP hydrolysis to translocate Ca$^{2+}$ from the cytosol back into the SR. When SLN binds to SERCA, it promotes Ca$^{2+}$ slippage without affecting the ATPase activity of SERCA. By doing so, SLN uncouples Ca$^{2+}$ transport activity from ATP hydrolysis of SERCA with concomitant heat generation. The increased duration of Ca$^{2+}$ transients in the cytosol also elevates mitochondrial Ca$^{2+}$, leading to enhanced mitochondrial oxidative metabolism and heat generation. (**E**) qPCR analysis of sarcolipin (*Sln*) expression in euploid and TcMAC21 mice fed a standard chow, high-fat diet (HFD), and HFD while housed at thermoneutrality (30°C). (**F**) Immunoblot of SLN and GAPDH (loading control) in muscle lysates of euploid, TcMAC21, and SLN overexpressing transgenic mice (*Sln*-Tg). (**G**) Immunoblot quantification of SLN using GAPDH as a loading control. The dotted line marks the euploid expression level. (**H–I**) Gastrocnemius immunofluorescent labeling of SLN-expressing muscle fibers. (**J**) Muscle fiber cross-sectional area (CSA) quantification from wheat germ agglutinin (WGA)-stained gastrocnemius. (**K**) Immunoblot of OXPHOS complex levels, with GAPDH as a loading control. (**L**) Quantification of OXPHOS complex levels relative to GAPDH. (**M–N**) Gastrocnemius immunofluorescent labeling of succinate dehydrogenase subunit B (SDHB)-expressing muscle fibers. (**O–P**) Seahorse respirometry analyses of frozen tissue samples. Shown here are the quadricep group average tracings for oxygen consumption rate (OCR) across the experimental time course. (**Q**) OCR of TcMAC21 mitochondrial complexes I, II, and IV relative to euploid, for 11 separate tissues (quadricep, Qd; extensor digitorum longus, EDL; gastrocnemius, Gast; plantaris, Plnt; soleus, Sol; tongue, Tng; brown adipose tissue, BAT; liver, Liv; heart, Hrt; inguinal white adipose tissue, iWAT; gonadal white adipose tissue, gWAT) and two dietary conditions (chow and high-fat diet [HFD]). DEGs, differentially expressed genes; PCGs, protein-coding genes; NPCGs, non-protein-coding genes. n=5 euploid and 4 TcMAC21 for RNA-sequencing experiments. n=8–9 euploid and 7–8 TcMAC21 for HFD qPCR. n=7–10 euploid and TcMAC21 for chow qPCR. n=4–5 euploid and TcMAC21 for HFD+thermoneutrality qPCR. n=6 euploid and 7 TcMAC21 for all immunoblots. n=7 euploid and 8 TcMAC21 for gastrocnemius CSA quantification. n=6 euploid and 4 TcMAC21 for all mitochondrial respiration assays; each biological replicate represents the average of three technical replicates.

The online version of this article includes the following source data and figure supplement(s) for figure 6:

**Source data 1.** Top left—Original uncropped membrane from imager showing all channels (red/green/blue), with the sarcolipin protein band labeled as SLN appearing blue.

**Source data 2.** Top left—Original uncropped membrane from imager showing all channels (red/green/blue), with the OXPHOS complex protein bands labeled as CI, CII, CIII, CIV, and CV appearing blue.

**Figure supplement 1.** The top up- and downregulated genes in skeletal muscle based on RNA-sequencing data.

**Figure supplement 2.** Gene expression analysis in skeletal muscle.

**Figure supplement 3.** Skeletal muscle cross-sectional area (CSA) analysis of MAC21 and euploid mice fed a high-fat diet.

**Figure supplement 4.** Mitochondrial respirometry analysis of gastrocnemius and quadricep muscle.

**Figure supplement 5.** Mitochondrial respirometry analysis of soleus and plantaris muscle.

**Figure supplement 6.** Mitochondrial respirometry analysis of extensor digitorum longus (EDL) and brown adipose tissue (BAT).

**Figure supplement 7.** Mitochondrial respirometry analysis of liver and tongue.

**Figure supplement 8.** Mitochondrial respirometry analysis of heart.

**Figure supplement 9.** Mitochondrial respirometry analysis of inguinal (iWAT) and gonadal (gWAT) white adipose tissue.

action, and general calcium handling (*Figure 6C* and *Figure 6—figure supplement 2*). While a few of the genes were significantly different in expression, the most notable upregulated gene with the biggest magnitude of change was *Sln* (*Figure 6C*), encoding the 31 amino acid single-pass membrane protein, SLN. SLN is a regulator and an uncoupler of the sarco(endo)plasmic reticulum calcium ATPase (SERCA) (*Bal and Periasamy, 2020*; *Wang et al., 2021*). Under normal circumstances in a resting muscle fiber, Ca$^{2+}$ cycling through the sarcoplasmic reticulum involves Ca$^{2+}$ leak through the ryanodine receptor (RYR) and its re-sequestration back to SR by the SERCA pump (*Barclay and Launikonis, 2022*; *Macdonald and Stephenson, 2001*; *Figure 6D*). Resting cytosolic Ca$^{2+}$ is further controlled by the store-operated calcium entry mechanism localized at the plasma membrane/T-tubule (*Pearce et al., 2022*). SERCA uses the energy derived from ATP hydrolysis to transport calcium from the cytosol back into the SR (*Toyoshima and Inesi, 2004*). When SLN binds to SERCA, it promotes Ca$^{2+}$ slippage without affecting the ATPase activity of the pump; therefore, more ATP needs to be consumed by the SERCA pump to re-sequester Ca$^{2+}$ back into SR in the presence of SLN. Effectively, SLN uncouples ATP hydrolysis from calcium transport into the SR (*Smith et al., 2002*; *Sahoo et al., 2013*). This results in futile SERCA pump activity, ATP hydrolysis, and heat generation (*Wang et al., 2021*; *Bal et al., 2012*; *Mall et al., 2006*). In addition, in the context of excitation-contraction coupling, increased duration of cytosolic calcium transients due to uncoupling of the SERCA pump can transiently elevate mitochondrial Ca$^{2+}$ and activates mitochondrial respiration and heat generation (*Barclay and Launikonis,*

2021; *Yi et al., 2011*), as well as calcium-dependent signaling that enhances oxidative metabolism in skeletal muscle (*Maurya et al., 2018*; *Maurya et al., 2015*; *Figure 6D*).

Interestingly, regardless of diet (chow or HFD) and temperature (22°C or 30°C), TcMAC21 mice consistently had ~20- to 30-fold upregulated *Sln* expression (*Figure 6E*). The expression of other calcium handling genes—phospholamban (*Pln*), calmodulin (*Calm2*), SERCA1a (*Atp2a1*), SERCA2a (*Atp2a2*)—were not significantly different between TcMAC21 and euploid mice. Likewise, overexpressing SLN in skeletal muscle of transgenic mice also did not alter SERCA expression (*Sopariwala et al., 2015*). The *Sln* gene is known to be upregulated in skeletal muscle by cold exposure (*Pant et al., 2015*; *Rowland et al., 2015a*; *Bal et al., 2016*). In contrast, the markedly upregulated *Sln* expression seen in TcMAC21 mice housed at ambient temperature (22°C) remained high even when the animals were housed at thermoneutrality (30°C). Consistent with the mRNA data, SLN protein levels were also strikingly upregulated in TcMAC21 skeletal muscle (gastrocnemius) (*Figure 6F–G*). Immunofluorescence also indicated substantially more SLN positive muscle fibers in TcMAC21 mice (*Figure 6H–I*). We included skeletal muscle lysate from SLN overexpression transgenic mouse (*Maurya et al., 2015*) as our positive control. While the transgenic mouse had ~2.5-fold higher level of SLN compared to controls, the SLN protein levels in TcMAC21 were ~4.5-fold higher than the euploid mice (*Figure 6G*). It is known that *Sln* can be induced in skeletal muscle as a compensatory response to muscle atrophy, dystrophy, and injury (*Chambers et al., 2022*). However, none of the genes involved in muscle repair, wasting, and atrophy were significantly upregulated (*Supplementary file 11*), suggesting that *Sln* overexpression is not due to structural or functional deficit of the skeletal muscle in TcMAC21 mice. Collectively, our data suggest that a mechanism exists that can achieve upregulation of endogenous mouse SLN at a level substantially higher than that seen in artificially overexpressed transgenic mice under non-pathological condition.

Quantification of histological sections revealed that the average gastrocnemius muscle fiber CSA in TcMAC21 was significantly smaller compared to euploid mice (*Figure 6J* and *Figure 6—figure supplement 3*). This suggests a shift from being a predominantly glycolytic muscle to a more oxidative muscle, as a smaller muscle fiber CSA is associated with a more oxidative muscle phenotype (*van Wessel et al., 2010*). In accordance, protein levels of mitochondrial complexes I–V were significantly higher in TcMAC21 gastrocnemius muscle compared to euploid controls (*Figure 6K–L*). Corroborating the Western blot data, immunofluorescence also showed substantially higher succinate dehydrogenase (SDHB) staining (marker of oxidative capacity) in gastrocnemius muscle of TcMAC21 mice (*Figure 6M–N*).

To determine if TcMAC21 mice have higher mitochondrial activity compared to euploid controls, we conducted mitochondrial respiration analyses in liver, BAT, iWAT, gWAT, and different muscle types (quadriceps, extensor digitorum longus, gastrocnemius, plantaris, soleus, tongue, and heart). Except soleus and heart, most muscle types from TcMAC21 fed either chow or HFD—regardless of whether they were predominantly slow-twitch (oxidative), fast-twitch (glycolytic), or mixed—showed significantly elevated oxygen consumption due to elevated mitochondrial respiration (*Figure 6O–Q* and *Figure 6—figure supplements 4–9*). Enhanced mitochondrial complex I, II, and IV activities were more pronounced in glycolytic muscle tissues, suggesting that these normally glycolytic muscle fibers may assume a more oxidative phenotype in TcMAC21 mice. The greater oxidative phenotype in glycolytic muscle tissues was not due to changes in fiber-type composition, as none of the fiber type-specific transcripts were different between genotypes (*Figure 6C*). These data are consistent with similar results reported for skeletal muscle-specific SLN overexpression mice (*Maurya et al., 2018*; *Maurya et al., 2015*; *Sopariwala et al., 2015*).

In contrast to muscle, mitochondrial activities of complexes I and II in interscapular BAT was largely unchanged in TcMAC21 (*Figure 6Q* and *Figure 6—figure supplement 6*), despite the pronounced differences in BAT histology and lipid content between TcMAC21 and euploid mice (*Figure 4K–L*). In chow-fed mice, however, mitochondrial complex IV activity was higher in BAT of TcMAC21. In subcutaneous (inguinal) white adipose tissue, mitochondrial respiration was significantly reduced in chow-fed TcMAC21 mice but unchanged in HFD-fed mice (*Figure 6—figure supplement 9*). In visceral (gonadal) white adipose tissue, mitochondrial respiration was also largely unchanged in either chow- or HFD-fed TcMAC21 mice (*Figure 6—figure supplement 9*). Unexpectedly and in striking contrast to muscle and BAT, mitochondrial complex I, II, and IV activity was markedly reduced in the liver and heart of both chow- and HFD-fed TcMAC21 compared to euploid mice (*Figure 6Q* and

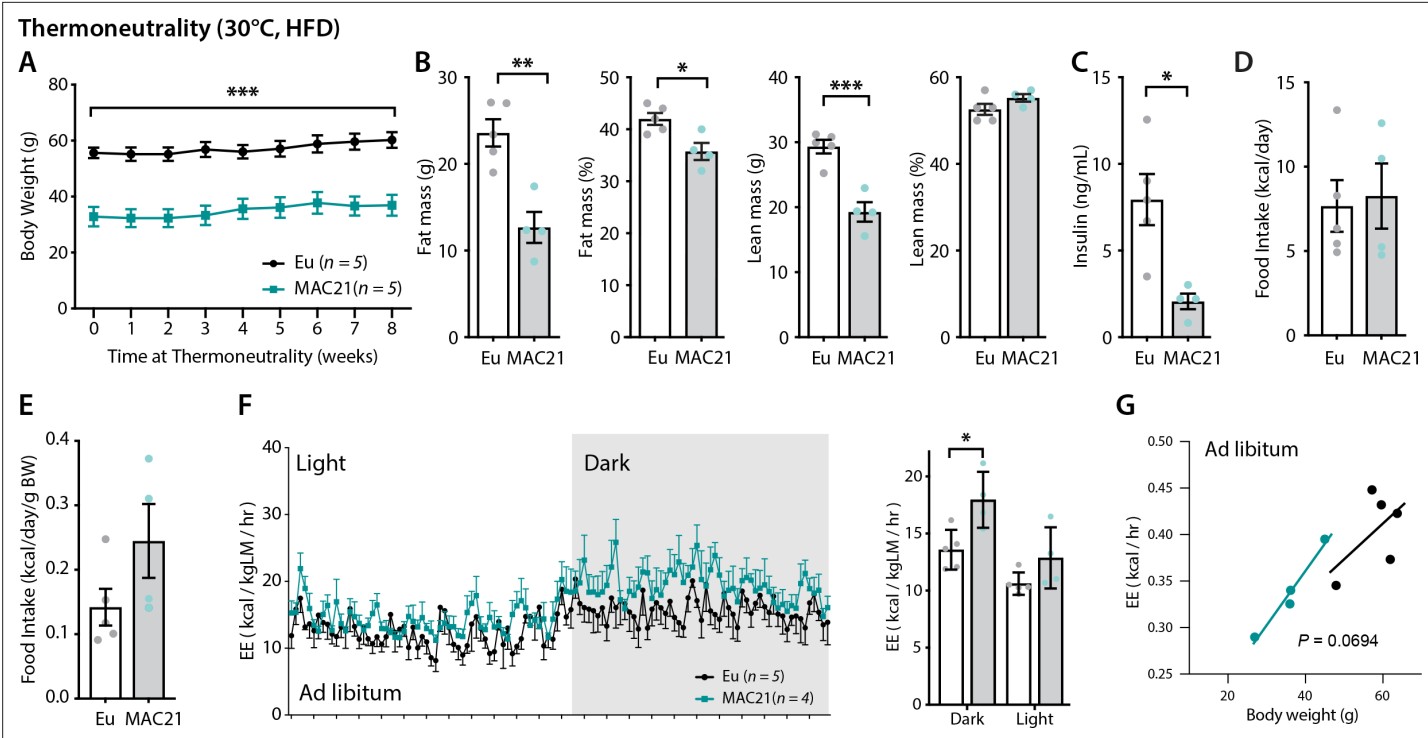

**Figure 7.** Increased energy expenditure (EE) persists in TcMAC21 mice housed at thermoneutrality. (**A**) Body weights of TcMAC21 mice and euploid controls housed at thermoneutrality (30°C) for 8 weeks. (**B**) Body composition analysis of absolute and relative (normalized to body weight) fat and lean mass. (**C**) Fasting insulin levels. (**D–E**) Absolute and relative (normalized to body weight) food intake. (**F**) EE in the dark and light cycles. *p<0.05 (two-way ANOVA). (**G**) EE as analyzed by ANCOVA where body weight was used as a covariate. Sample size for all data: euploid (n=5) and TcMAC21 (n=4–5).

The online version of this article includes the following figure supplement(s) for figure 7:

**Figure supplement 1.** Histology of high-fat diet (HFD)-fed euploid and MAC21 mice housed at thermoneutrality (30°C).

*Figure 6—figure supplements 7–8*), despite the liver of TcMAC21 mice being significantly 'healthier' with much reduced lipid accumulation (*Figures 2I and 3E*).

All together, these data suggest that SLN-mediated futile SERCA activity in skeletal muscle is a potential driver underlying the hypermetabolism phenotypes seen in TcMAC21 mice. Because the futile SERCA pump activity consumes more ATP to transport $Ca^{2+}$ into SR due to $Ca^{2+}$ slippage (*Wang et al., 2021*; *Smith et al., 2002*; *Sahoo et al., 2013*; *Mall et al., 2006*; *Sahoo et al., 2015*), this creates a huge energy demand and markedly drives up the activity of mitochondrial respiration to supply the ATP needed for the SERCA pump. Presumably, this effect results in systemic channeling of metabolic substrates (e.g. lipids) into skeletal muscle to fuel the elevated mitochondrial respiration rate. This in turn leads to secondary improvements and protection—reduced fat accumulation and smaller adipocytes—we observed in liver, BAT, gWAT, and iWAT despite minimal changes in the transcriptomes of these tissues.

## Potential regulators of SLN expression in skeletal muscle

Having established that SLN-mediated futile SERCA activity underlies hypermetabolism in TcMAC21 mice, we next asked what regulator(s) promote the upregulation of endogenous SLN, as little is known about what controls its expression in skeletal muscle. We took advantage of the observation that SLN overexpression appears to be locked in the 'on' position in skeletal muscle and could not be downregulated to control levels at thermoneutrality (*Figure 6E*). Thyroid hormone ($T_3$) is likely not the driver of SLN overexpression since serum $T_3$ was not different between TcMAC21 and euploid housed at thermoneutrality (*Figure 2—figure supplement 1*); in addition, SLN was shown to be upregulated in hypothyroid mice compared to euthyroid controls (*Kaspari et al., 2020*). When housed at thermoneutrality (30°C), TcMAC21 mice consumed a similar amount of calories as the euploid controls but maintained their lean phenotypes with lower fasting insulin levels (*Figure 7A–E*). When normalized

to lean mass, TcMAC21 mice appeared to have higher EE in the dark cycle (*Figure 7F* and *Supplementary file 12*). However, when body weight was used as a covariate in ANCOVA, the higher EE seen in TcMAC21 was short of significance (p=0.0693), and this could be due to the small sample size (*Figure 7G*). Consistent with lower adiposity, TcMAC21 mice housed at thermoneutrality had smaller adipocyte cell size and lower lipid contents in BAT and liver (*Figure 7—figure supplement 1*). Tissue collection at termination of the study also showed significantly smaller fat mass and reduced BAT and liver weight in TcMAC21 mice (*Supplementary file 13*).

From these data we surmised that the regulator(s) of SLN would likewise remain unchanged (i.e. continued to be up- or down-regulated) at thermoneutrality. We therefore performed RNA-sequencing on skeletal muscle isolated from HFD-fed TcMAC21 housed at thermoneutrality. RNA-sequencing data showed a total of 113 DEGs in the skeletal muscle of TcMAC21 mice, with 59 being upregulated and 54 being downregulated (*Figure 8A* and *Supplementary file 14*). Of the 59 upregulated genes, 56 were PCGs and 3 were NPCGs; the downregulated group had 51 PCGs and 3 NPCGs. Next, we compared the skeletal muscle transcriptomes of euploid vs TcMAC21 mice housed at 22°C with the transcriptomes of euploid vs TcMAC21 mice housed at 30°C. This tells us which genes are differentially expressed by TcMAC21 mice relative to euploid controls at both temperatures. There were 848 genes differentially expressed by only TcMAC21 mice housed at 22°C. This includes 392 upregulated (299 PCGs and 93 NPCGs) and 456 downregulated (386 PCGs and 70 NPCGs) genes (*Figure 8B*). There were 31 genes differentially expressed by only TcMAC21 mice housed at 30°C, with 22 upregulated (19 PCGs and 3 NPCGs) and 9 downregulated (7 PCGs and 2 NPCGs) genes. Notably, there were 81 genes differentially expressed by TcMAC21 at both temperatures, with 37 upregulated (all PCGs) and 44 downregulated (43 PCGs and 1 NPCG) genes.

To further narrow our candidate *Sln* regulators, we directly compared the expression of mouse genes in the skeletal muscle of TcMAC21 mice housed at 22°C and 30°C. Because *Sln* expression was similarly upregulated in the skeletal muscle of TcMAC21 mice housed at 22°C and 30°C, we presumed the regulator(s) of *Sln* would also display minimal variation across both temperatures. We therefore focused on genes that remained unchanged at 22°C and 30°C in TcMAC21 skeletal muscle. We first filtered out all genes above or below Log2(FC) of 0.5 or –0.5, which left us with 3754 transcripts whose expression remained relatively unchanged at both temperatures. We next sorted the data by p-value, then compared the entire list of unchanged transcripts to the list of shared DEGs shown in *Figure 8B*, and found that only five genes were present in both (*Figure 8C*). This list contains one upregulated gene (A930018M24Rik) and four downregulated PCGs (*Csde1*, *Kmt5a*, *Vasp*, and *Calm1*). Again, these are the genes that are differentially expressed by TcMAC21 mice housed at 22°C and 30°C as compared to euploid, which remain unchanged in TcMAC21 mice across temperature. Thus, these are potential *Sln* regulators that are encoded in the mouse genome.

Alternatively, the regulator(s) of *Sln* expression could be one or more Hsa21-drived human genes. There was a total of 106 human transcripts expressed by TcMAC21 mice housed at 22°C and 30°C, with 94 PCGs and 12 NPCGs (*Figure 8D*). There were four human PCGs only expressed in TcMAC21 mice housed at 30°C and 83 human genes whose expression was turned off at 30°C. Interestingly, a majority (75 out of 83) of the human genes that were turned off at 30°C are NPCGs. The human genes that we postulated as most likely to regulate *Sln* expression would be those that change the least at 22°C and 30°C. We first calculated the difference in Log2(FC) values of all human genes expressed at 22°C and 30°C in TcMAC21 skeletal muscle (i.e. Log2(FC at 22°C) - Log2(FC at 30°C)). Those values closest to zero represent genes with the most consistent expression across both housing temperatures. Genes with a negative Log2(FC) difference are expressed at higher levels at 30°C, and genes with a positive Log2(FC) difference are expressed at higher levels at 22°C. The ~20 genes with the closest-to-zero Log2(FC) difference are listed in yellow (*Figure 8E*), and these would be considered potential regulators of *Sln* derived from Hsa21. Altogether, our analysis has yielded a defined number of potential *Sln* regulators, from the mouse or human genome, that can be experimentally tested (either singly or in combination) in follow-up studies.

## Discussion

While assessing the systemic metabolic impact of cross-species aneuploidy (a human chromosome in mice), we made several striking and unexpected discoveries. Given that TcMAC21 mice recapitulate multiple features of human trisomy 21—most notably structural and functional neurological deficits

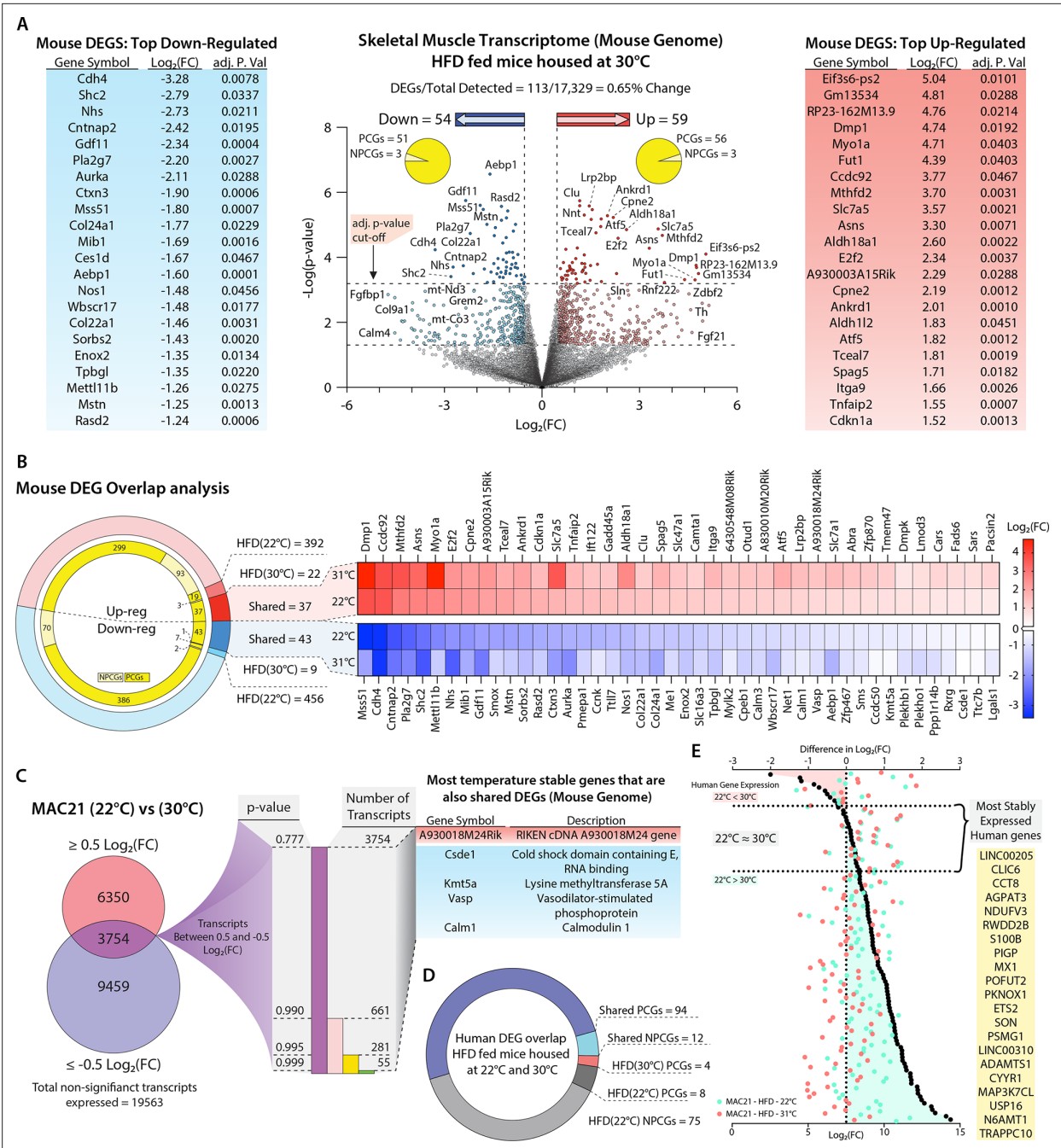

**Figure 8.** Potential regulators of sarcolipin expression in skeletal muscle. (**A**) Volcano plot of skeletal muscle transcriptome (transcripts from the mouse genome only). The lower dotted line denotes significance at the p-value cut-off (p=0.05), and the upper dotted line denotes significance at the adjusted p-value cut-off (adj. p=0.05). The vertical dotted lines denote a Log2(fold change [FC]) of –0.5 or 0.5. Flanking the volcano plot are the top down- and upregulated mouse genes. (**B**) Comparison of differentially expressed genes (DEGs) shared and not shared by TcMAC21 mice housed at 22°C vs. 30°C. Heat map showing all significantly up- or downregulated shared genes. (**C**) Direct comparison of TcMAC21 mice housed at 22°C and 30°C for most stably expressed mouse genes. Data filtered first by genes with Log2(FC) within ±0.5 (least change), then by lowest significance, and finally compared to the shared DEGs found in B. Table shows DEGs with least variation in expression at 22°C and 30°C. (**D**) Overlap analysis of Hsa21-derived human transcripts expressed in TcMAC21 mice (skeletal muscle) housed at 22°C and 30°C. (**E**) Graph showing the most stably expressed Hsa21-derived human genes in the TcMAC21 gastrocnemius. Top axis refers to the difference in Log2(FC) between human genes in TcMAC21 mice housed at 22°C and 30°C (data represented as black dots). Bottom axis shows the Log2(FC) value of a particular gene at 22°C and 30°C (data represented as a light blue-green or light red dot, respectively). The gene list shows all the human genes expressed by both groups with the least amount of change in expression between the two temperatures.

and craniofacial skeletal alterations (*Kazuki et al., 2020*) and also a small body size compared to their euploid littermates—we initially expected the animals to show metabolic changes such as obesity and glucose intolerance reminiscent of the DS population (*Bertapelli et al., 2016*) or the metabolic dysregulation (e.g. insulin resistance and dyslipidemia) seen in the Ts65Dn DS mouse model (*Sarver et al., 2023*). Instead, the TcMAC21 mice have several hallmarks of hypermetabolism. They are lean despite greater caloric intake, and have markedly elevated oxidative metabolic rate as indicated by increased VO$_2$, EE, mitochondrial respiration in skeletal muscle, and by a shift to a more oxidative phenotype in skeletal muscle.

Although the higher prevalence of obesity and diabetes is well documented in the DS population (*Bertapelli et al., 2016*), the mechanism of how gene dosage imbalance causes metabolic dysregulation in vivo is largely unknown and limited to only a few studies (*Dierssen et al., 2020*; *Fructuoso et al., 2018*). Until our recent study (*Sarver et al., 2023*), the scope of the previous metabolic studies using DS mouse models was limited. Our present study represents one of the most comprehensive metabolic analyses carried out in DS mouse models to date. To our surprise, TcMAC21 phenotypes appear to be the opposite of what is seen in DS. We can speculate on possible reasons for this: (1) The strong physiological effects of SLN overexpression in the skeletal muscle of TcMAC21 override and mask potential deleterious effects of high-fat feeding seen in the euploid controls. It is important to note that SLN overexpression is not seen in other commonly used DS mouse models such as the Ts65Dn (*Sarver et al., 2023*). Although candidate transcriptional drivers of *Sln* overexpression have been identified (*Figure 8*), it remains to be determined which among these are responsible for *Sln* overexpression in the skeletal muscle of TcMAC21 mice. (2) The unexpected phenotypes of TcMAC21, including SLN overexpression, could be the result of complex interactions between human genes and the mouse genome that we do not fully understand. The human Chr21 also contains a significant number of non-coding human genes (>400) with uncertain effects on the mouse transcriptome. Further, the human proteins may alter the stoichiometric compositions of multi-protein complexes in an unpredictable way. For this reason, and for comparison, it is imperative to carry out similar comprehensive metabolic studies in other established DS mouse models (e.g. Dp(16)1Yey/+ and Ts66Yah) where all the trisomic genes are derived from the mouse instead of human, and also without extra trisomic genes not found in Hsa21 (*Li et al., 2007*; *Duchon et al., 2022*). It is possible the trisomic mouse orthologs of Hsa21 genes are overexpressed and interact with the rest of the mouse genome in a manner that more closely reflects what might be seen in the adipose tissue, liver, and skeletal muscle of DS. This is an important issue to resolve in terms of selecting appropriate DS mouse models to best reflect the metabolic phenotypes seen in DS. While TcMAC21 is considered the best *genetic* model for DS in mice, the mouse apparently has a means of overcoming the negative metabolic impact of trisomy. Uncovering this mechanism could be important for the treatment or prevention of obesity and diabetes.

What could be the mechanism underlying the striking metabolic phenotypes of TcMAC21? One potential driver of hypermetabolism appears to be SLN overexpression in the skeletal muscle of TcMAC21 mice, leading to persistent uncoupling of the SERCA pumps, heat generation, and energy dissipation. This mechanism could explain multiple unexpected observations: despite a marked reduction in lipid accumulation in the liver, BAT, and WAT, surprisingly few changes were seen in the transcriptomes of these tissues, and none could account for the striking tissue histology. Importantly, none of the key genes (e.g. *Ucp1* in BAT and browning/beiging genes in iWAT) involved in thermogenesis and lipid metabolism were significantly different between TcMAC21 and euploid mice, and yet the mice had elevated deep colon temperature. While the metabolic activity (i.e. mitochondrial respiration) of BAT was largely unchanged, complex I, II, and IV activities in the liver were surprisingly downregulated. Mitochondrial respiration was either downregulated or unchanged in iWAT and gWAT. All these striking phenotypes could be explained by metabolic substrates (e.g. lipid) channeling away from liver, BAT, and WAT, and into skeletal muscle to fuel elevated mitochondrial respiration, as this was needed to meet the high ATP demand created by futile SERCA pump activity.

We largely ruled out the involvement of thyroid hormones in promoting hypermetabolism in TcMAC21 mice. Under the basal state when mice were fed a standard chow, or HFD-fed mice housed at thermoneutrality, circulating T$_3$ levels were not different between groups despite the hyperactivity and hypermetabolism of TcMAC21 mice. Although serum T$_3$ levels were elevated in HFD-fed TcMAC21 housed at room temperature (RT) (22°C), the predicted T$_3$ effects were largely absent. It is known that

$T_3$ negatively regulates the expression of SLN, as hypothyroidic mice have significantly upregulated SLN mRNA and protein in skeletal muscle compared to euthyroid mice (*Kaspari et al., 2020*). Instead of the expected downregulation, SLN expression was dramatically elevated in the skeletal muscle of TcMAC21 mice. Through its nuclear hormone receptor, $T_3$ regulates many metabolic genes in adipose tissue, liver, and skeletal muscle (*Mullur et al., 2014*), and yet none of the well-known $T_3$-regulated genes were upregulated in these tissues despite elevated $T_3$ levels. In addition, $T_3$ is a potent inducer of *Ucp1* expression in BAT (*Yau et al., 2019*) and white adipose tissue beiging (*Johann et al., 2019*), but *Ucp1* transcript levels were not upregulated in BAT or the white adipose tissue of TcMAC21 mice. This apparent $T_3$ resistance in HFD-fed TcMAC21 mice is likely a compensatory response to an already elevated body temperature and the hypermetabolism induced by SLN-mediated futile SERCA activity and heat generation in skeletal muscle. If $T_3$ further increased the metabolic rate of TcMAC21 mice, this would likely overtax the heat dissipation mechanism leading to hyperthermia and possible death.

By collapsing the proton motive force within the intermembrane space of mitochondria, UCP1-mediated mitochondrial uncoupling in BAT could raise body temperature and enhance EE (*Chouchani et al., 2019*). Although thermal imaging highlighted increased skin temperature around the interscapular BAT region of TcMAC21 mice, the thermal signal could be produced by local muscle around the interscapular area. Our gene expression, BAT transcriptomics, and functional data, however, do not support UCP1-mediated uncoupling in BAT as a probable mechanism for the hypermetabolism and elevated thermogenesis seen in TcMAC21 mice. Neither the expression of *Ucp1* nor mitochondrial activity was consistently elevated in the BAT of TcMAC21 mice fed chow or HFD. Housing mice at a thermoneutral temperature would suppress BAT activity and beiging in iWAT (*Gonzalez-Hurtado et al., 2018*), yet TcMAC21 mice housed at thermoneutrality for an extended period (>2 months) retained elevated EE and lean body weight comparable to TcMAC21 mice housed at 22°C. Together, these data argue against BAT being a significant contributor to the thermogenesis seen in TcMAC21 mice.

Recent studies have also shown that creatine-driven futile substrate cycling and SERCA2b-mediated calcium cycling in beige fat can promote EE and thermogenesis (*Kazak et al., 2015*; *Ikeda et al., 2017*). Regardless of housing temperatures, none of the genes involved in creatine/phosphocreatine futile cycle (e.g. *Slc6a8*, *Gatm*, *Gamt*, *Ckmt1*, *Ckmt2*) or the SERCA2b-RyR2 pathway (*Atp2a2*, *Ryr2*) were upregulated in iWAT, suggesting that these pathways are likely not involved in the hypermetabolic phenotypes of TcMAC21. Promoting metabolic inefficiency in skeletal muscle by UCP1 or UCP3 overexpression can increase metabolic rate and prevent diet-induced obesity (*Clapham et al., 2000*; *Son et al., 2004*; *Li et al., 2000*). In the skeletal muscle of TcMAC21 mice, *Ucp1* and *Ucp3* transcripts were not upregulated, suggesting that mitochondrial uncoupling by UCP1 or UCP3 is also unlikely to contribute to the observed lean and hypermetabolic phenotypes.

SLN plays a role in modulating body weight by promoting EE through its effects on the SERCA pump (*Bal et al., 2012*; *Maurya et al., 2015*; *Bombardier et al., 2013*; *MacPherson et al., 2016*). Binding of SLN to SERCA does not interfere with ATP hydrolysis of the pump, but reduces calcium transport into the SR lumen through a calcium 'slippage' mechanism (*Wang et al., 2021*; *Sahoo et al., 2013*; *Mall et al., 2006*; *Sahoo et al., 2015*). Thus, SLN promotes futile cycling of the SERCA pumps, ATP hydrolysis, and heat generation. Consistent with this model, mice lacking SLN are cold intolerant and gain more weight on an HFD due to reduced EE (*Bal et al., 2012*; *Bombardier et al., 2013*; *MacPherson et al., 2016*); conversely, transgenic overexpression of SLN in skeletal muscle promotes a lean phenotype (*Maurya et al., 2015*).

Under normal situations, SLN is mainly expressed in slow-twitch oxidative muscle tissues such as the soleus and diaphragm, with little expression in fast-twitch glycolytic muscle tissues such as the gastrocnemius and quadriceps (*Babu et al., 2007*). In the case of hypothyroid mice, endogenous SLN is only markedly upregulated in oxidative muscle fibers (soleus and diaphragm) but not the large glycolytic muscle tissues (*Kaspari et al., 2020*). It has been argued that in rodents the futile SERCA activity and EE occurring in soleus and diaphragm (both are small in size) is quantitatively insufficient to effect significant changes in body weight (*Campbell and Dicke, 2018*). Notwithstanding the artificial overexpression of SLN in both fast-twitch glycolytic and slow-twitch oxidative muscle fibers (*Maurya et al., 2015*), it is unclear whether endogenous SLN can be substantially induced in large glycolytic muscle tissues such as the gastrocnemius and quadriceps under non-pathological conditions. Our data helps to resolve this issue. In TcMAC21 mice, endogenous SLN is markedly upregulated in a largely

glycolytic muscle (gastrocnemius), suggesting that a mechanism indeed exists that can substantially elevate SLN transcript and protein in muscle fiber types that normally have low expression. Corroborating this is the observation that mitochondrial respiration is significantly elevated in most muscle types—gastrocnemius (fast-twitch), quadriceps (fast-twitch), plantaris (mixed), and extensor digitorum longus (mixed).

The huge ATP demand created by the futile SERCA pump activity appears to be the cause of upregulated protein expression of complexes I-IV, elevated mitochondrial respiration, and the assumption of an oxidative phenotype in muscle tissues that are normally glycolytic. This could also reflect skeletal muscle adaptation to increased physical activity level seen in TcMAC21 mice. This reprogramming of glycolytic muscle tissues into an oxidative phenotype is likely due to increased duration of calcium transients (*Baylor and Hollingworth, 2003*), leading to calcium-dependent activation of mitochondrial respiration and calcium-dependent signaling that drives increased production of OXPHOS proteins (*Maurya et al., 2018*; *Maurya et al., 2015*; *Robb-Gaspers et al., 1998*; *Jouaville et al., 1999*; *Tarasov et al., 2012*; *Glancy et al., 2013*). Since skeletal muscle makes up ~40% of the body's weight and is a major energy consuming tissue (*Rolfe and Brown, 1997*; *Zurlo et al., 1990*), elevated EE due to a combination of hyperactivity and futile SERCA pump activity—especially the big muscles (gastrocnemius and quadriceps)—will have substantial impact on total energy balance and body weight. As an added benefit, increased fuel consumption for non-shivering thermogenesis driven by the futile SERCA pump would channel lipid substrates into skeletal muscle for oxidation and prevent excess lipid accumulation in liver, BAT, and white adipose tissue, thus conferring an overall healthy metabolic profile with preserved systemic insulin sensitivity.

We do not know the relative contribution of hyperactivity and futile SERCA activity to the overall lean phenotype of the TcMAC21 mice. Future studies using a TcMAC21 mouse model specifically lacking SLN will help answer the important question of whether SLN indeed plays a causal role in promoting the lean phenotype we observed in TcMAC21 mice. Although TcMAC21 mice have increased voluntary physical activity, which can elevate metabolism, our data suggest that futile SERCA activity in skeletal muscle could be another potential driver of the hypermetabolic phenotype. Overexpression of SLN in skeletal muscle does not significantly alter voluntary physical activity at ambient RT or at thermoneutrality, yet the SLN transgenic mice consistently gained significantly less weight in response to high-fat feeding (*Maurya et al., 2015*; *Bombardier et al., 2013*). Further, SLN appears to be required for effective adiposity reduction in HFD-fed mice with access to a running wheel (*Gamu et al., 2015*). Elevated locomotor activity in TcMAC21 mice, however, would likely amplify the effect of SLN-mediated uncoupling of the SERCA pumps, as muscle movement results in greater SR $Ca^{2+}$ cycling. In cultured myotubes, caffeine-induced release of $Ca^{2+}$ from the SR via the ryanodine receptor/channel (RYR1) would only activate calcium-dependent signaling in the presence, but not absence, of SLN; conversely, inhibiting $Ca^{2+}$ release through RYR1 using dantrolene blocks calcium-dependent signaling only in SLN-expressing, but not SLN-deficient, myotubes (*Maurya et al., 2018*). Thus, increased SR $Ca^{2+}$ cycling due to elevated physical activity would promote EE and also create a favorable cellular context to amplify the effects of SLN overexpression on uncoupling of the SERCA pumps. Accordingly, EE and the degree of leanness seen in TcMAC21 mice are significantly greater in magnitude than the SLN transgenic mice.

To fully harness SLN-mediated futile SERCA activity for energy dissipation and weight loss requires an understanding of how endogenous SLN expression is regulated, but the mechanism of which remains largely unknown. TcMAC21 mice offer critical insights into the switches that control SLN expression in skeletal muscle. Regardless of temperatures (ambient or thermoneutral), elevated *Sln* expression in TcMAC21 appeared to be in the 'on' position and could not be turned off. We exploited this information to zoom in on a defined set of mouse and Hsa21-derived transcripts—some of which are transcription factors (e.g. *PKNOX1*, *ETS2*), RNA binding proteins (e.g. *Csde1*), and epigenetic regulators (e.g. *Kmt5a*)—that remain significantly up- or down-regulated in skeletal muscle at both 22°C and 30°C, and show <1% variability across the two temperatures. We presume that one or more of these genes could potentially be the sought after 'switches' of *Sln* expression. Additional studies are needed to identify such critical on/off switches. A recent study has suggested that the C-terminal cleavage fragment of TUG can upregulate *Sln* expression in the skeletal muscle when it enters the nucleus and forms a complex with PPAR-γ and PGC-1α (*Habtemichael et al., 2021*). However, this

pathway is largely abrogated in mice fed an HFD; thus, we think this transcriptional mechanism is likely not involved in upregulating *Sln* expression in TcMAC21 mice fed an HFD.

Weight loss can be achieved by reducing caloric intake or promoting EE, and the latter can be physiologically accomplished by increasing physical activity (e.g. exercise) or through shivering and non-shivering thermogenesis (*Chouchani et al., 2019*; *Roesler and Kazak, 2020*). SLN-mediated non-shivering thermogenesis offers an attractive approach for promoting EE and weight loss: Expression of SLN is largely restricted to the striated muscle of all mammals, including humans (*Paran et al., 2015*; *Rowland et al., 2015b*; *Vangheluwe et al., 2005*; *Fajardo et al., 2013*). Compared to BAT (~1.5% of body weight in young men) (*Leitner et al., 2017*), the total mass of skeletal muscle (~40% of body weight) makes futile SERCA activity in this large tissue especially effective in energy dissipation (*Smith et al., 2013*). Importantly, overexpression of SLN in skeletal muscle does not appear to have adverse effects; rather, SLN transgenic mice have higher endurance capacity and improved muscle performance due to enhanced oxidative capacity (*Sopariwala et al., 2015*). In summary, our work provides further proof-of-concept that endogenous SLN-mediated uncoupling of SERCA pumps to enhance EE can potentially be harnessed for systemic metabolic health.

## Materials and methods
### Mouse model
The transchromosomic TcMAC21 (simply referred to as MAC21) mice carrying a near-complete human chromosome 21 were generated and genotyped as previously described (*Kazuki et al., 2020*). Because MAC21 male mice are infertile, all the female mice were used for breeding. Hence, metabolic analyses were conducted on male mice only. MAC21 mice were initially maintained on an outbred background (ICR strain mice). After crossing for eight generations onto BDF1 (C57BL/6J (B6) × DBA/2J (D2)), MAC21 were transferred to the Riken Animal Resource (BRC No. RBRC05796 and STOCK Tc (HSA21q-MAC1)). Characterizations of MAC21 were performed on mice (75% B6/25% D2 on average) produced by crossing B6 males with trisomic B6D2 females (*Kazuki et al., 2020*). Euploid littermates were used as controls throughout the studies.

Mice were fed a standard chow (Envigo; 2018SX) or HFD (60% kcal derived from fat, #D12492, Research Diets, New Brunswick, NJ, USA). Mice were housed in polycarbonate cages on a 12 hr:12 hr light-dark photocycle with ad libitum access to water and food. HFD was provided for 16 weeks, beginning at 5 months of age. At termination of the study, all mice were fasted for 2 hr and euthanized. Tissues were collected, snap-frozen in liquid nitrogen, and kept at 80°C until analysis.

For thermoneutral studies, mice were housed in a temperature-regulated (ambient temp. maintained at 30 ± 1°C) animal facility at Johns Hopkins University School of Medicine. Mice were acclimatized for 2 full weeks prior to experimentation. All experimental procedures conducted on these mice (i.e. GTT and ITT, fasting-refeeding blood collections, tissue dissections, etc.) were completed within the 30°C housing unit to prevent temperature fluctuation.

All mouse protocols were approved by the Institutional Animal Care and Use Committee of the Johns Hopkins University School of Medicine (animal protocol # MO19M481). All animal experiments were conducted in accordance with the National Institute of Health guidelines and followed the standards established by the Animal Welfare Acts.

### Body composition analysis
Body composition analyses for total fat and lean mass, and water content were determined using a quantitative magnetic resonance instrument (Echo-MRI-100, Echo Medical Systems, Waco, TX, USA) at the Mouse Phenotyping Core facility at Johns Hopkins University School of Medicine.

### Complete blood count analysis
A complete blood count on blood samples was performed at the Pathology Phenotyping Core at Johns Hopkins University School of Medicine. Tail vein blood was collected using EDTA-coated blood collection tubes (Sarstedt, Nümbrecht, Germany) and analyzed using Procyte Dx analyzer (IDEXX Laboratories, Westbrook, ME, USA).

### Indirect calorimetry
Chow or HFD-fed MAC21 male mice and euploid littermates were used for simultaneous assessments of daily body weight change, food intake (corrected for spillage), physical activity, and whole-body

metabolic profile in an open flow indirect calorimeter (Comprehensive Laboratory Animal Monitoring System, CLAMS; Columbus Instruments, Columbus, OH, USA) as previously described (*Sarver et al., 2020*). In brief, data were collected for 3 days to confirm mice were acclimatized to the calorimetry chambers (indicated by stable body weights, food intakes, and diurnal metabolic patterns), and data were analyzed from the subsequent 3 days. Mice were observed with ad libitum access to food, throughout the fasting process, and in response to refeeding. Rates of oxygen consumption ($\dot{V}_{O2}$; mL·kg$^{-1}$·h$^{-1}$) and carbon dioxide production ($\dot{V}_{CO2}$; mL·kg$^{-1}$·h$^{-1}$) in each chamber were measured every 24 min. Respiratory exchange ratio (RER = $\dot{V}_{CO2}$/ $\dot{V}_{O2}$) was calculated by CLAMS software (version 4.93) to estimate relative oxidation of carbohydrates (RER = 1.0) versus fats (RER = 0.7), not accounting for protein oxidation. EE was calculated as EE = $\dot{V}_{O2}$× [3.815 + (1.232×RER)] and normalized to lean mass. In addition, EE was also analyzed using ANCOVA (*Tschöp et al., 2012*). Physical activities (total and ambulatory) were measured by infrared beam breaks in the metabolic chamber.

### Fecal bomb calorimetry and assessment of fecal parameters
Fecal pellet frequency and average fecal pellet weight were monitored by housing each mouse singly in clean cages for 3 days and counting the number of fecal pellets and recording their weight at the end of each 24 hr period. The average of the 3 days was used to generate a mouse average, which were then averaged within a group for comparison across genotype. Fecal pellets from 3 days were combined and shipped to the University of Michigan Animal Phenotyping Core for fecal bomb calorimetry. Briefly, fecal samples were dried overnight at 50°C prior to weighing and grinding them to powder. Each sample was mixed with wheat flour (90% wheat flour, 10% sample) and formed into 1.0 g pellet, which was then secured into the firing platform and surrounded by 100% oxygen. The bomb was lowered into water reservoir and ignited to release heat into the surrounding water. Together these data were used to calculate fecal pellet frequency (bowel movements/day), average fecal pellet weight (g/bowel movement), fecal energy (cal/g feces), and total fecal energy (kcal/day).

### Thermography tests
Deep colonic temperature was measured by inserting a lubricated (Medline, water soluble lubricating jelly, MDS032280) probe (Physitemp, BAT-12 Microprobe Thermometer) into the anus at a depth of 2 cm. Stable numbers were recorded on 3 separate days in both the dark and light cycle for each mouse. Skin temperature measurements and images of the tail, abdomen, and suprascapular regions of mice were taken across 3 days, in both the dark and light cycle, using a thermal imaging camera (Teledyne FLIR, Sweden, FLIR-C2) set at a constant distance of ~16 inches away from the specimen.

### Bone length measurements
Tibias were dissected and excess tissue was cleared away. A Mitutoyo Corp. digital caliper (500-196-30) was used to measure end-to-end bone length.

### GTT and ITT
For GTTs, mice were fasted for 6 hr before glucose injection. Glucose (Sigma, St. Louis, MO, USA) was reconstituted in saline (0.9 g NaCl/L), sterile-filtered, and injected intraperitoneally (i.p.) at 1 mg/g body weight. Blood glucose was measured at 0, 15, 30, 60, and 120 min after glucose injection using a glucometer (NovaMax Plus, Billerica, MA, USA). Blood was collected at 0, 15, and 30 min time points for serum isolation followed by insulin ELISA. For ITTs, food was removed 2 hr before insulin injection. Insulin was diluted in saline, sterile-filtered, and injected i.p. at 1.0 U/kg body weight. Blood glucose was measured at 0, 15, 30, 60, and 90 min after insulin injection using a glucometer (NovaMax Plus).

### Fasting-refeeding insulin tests
Mice fasted overnight (~16 hr) then reintroduced to food. Blood glucose was monitored at the 16 hr fast time point (time = 0 hr refed) and at 1, 2, and 3 hr into the refeeding process. Blood was collected at the 16 hr fast and 2 hr refed time points for insulin ELISA. HOMA-IR was calculated as follows (*Matthews et al., 1985*): [fasting insulin (uIU/mL) × blood glucose (mmol/L)]/22.5.

### Blood and tissue chemistry analysis
Tail vein blood samples were allowed to clot on ice and then centrifuged for 10 min at 10,000 × *g*. Serum samples were stored at –80°C until analyzed. Serum triglycerides and cholesterol were

measured according to the manufacturer's instructions using an Infinity kit (Thermo Fisher Scientific, Middletown, VA, USA). NEFA were measured using a Wako kit (Wako Chemicals, Richmond, VA, USA). Serum β-hydroxybutyrate (ketone) concentrations were measured with a StanBio Liquicolor kit (StanBio Laboratory, Boerne, TX, USA). Serum insulin (Crystal Chem, 90080), adiponectin (MilliporeSigma, EZMADP-60K), leptin (MilliporeSigma, EZML-82K), growth hormone (MilliporeSigma, EZRMGH-45K), IGF-1 (Crystal Chem, 80574), T3 (Calbiotech, T3043T-100), and T4 (Calbiotech, T4044T-100) levels were measured by ELISA according to the manufacturer's instructions. Pancreatic insulin and SST were isolated from pancreas samples using acid-ethanol extraction. Briefly, pancreatic samples were in a solution of acid-ethanol (1.5% HCl in 70% EtOH) and incubated overnight at –20°C. They were then homogenized and incubated for an additional night at –20°C. The following day, samples were centrifuged at 4°C to pellet debris. The aqueous protein-rich solution was transferred to a new tube and neutralized with 1 M Tris pH 7.5. Prior to insulin quantification by ELISA, samples were diluted 1:1000 with ELISA sample diluent. Following ELISA quantification of pancreatic insulin (Mercodia, 10-1247-01) and SST (Phoenix Pharmaceuticals Inc, EK-060-03) protein concentrations of pancreatic samples were quantified using a Bradford assay (Sigma-Aldrich, B6916). Insulin and SST values were then normalized to protein concentration, averaged, and compared across groups.

## Histology and quantification

Inguinal (subcutaneous) white adipose tissue (iWAT), gonadal (visceral) white adipose tissue (gWAT), liver, pancreas, suprascapular BAT, and gastrocnemius muscle were dissected and fixed in formalin. Paraffin embedding, tissue sectioning, and staining with hematoxylin and eosin were performed at the Pathology Core Facility at Johns Hopkins University School of Medicine. Images were captured with a Keyence BZ-X700 All-in-One fluorescence microscope (Keyence Corp., Itasca, IL, USA). Adipocyte (gWAT and iWAT) and gastrocnemius CSA, as well as the total area covered by lipid droplets in hepatocytes and BAT adipocytes, were measured on hematoxylin and eosin-stained slides using ImageJ software (*Schneider et al., 2012*). For CSA measurements, all cells in one field of view at 100× magnification per tissue section per mouse were analyzed. For iWAT and gWAT adipocyte CSA, at least 150 cells were quantified per mouse. For pancreatic β-islet CSA, 40× magnification images were stitched together from sections at two depths of the pancreas for a single mouse. All β-islets across the two stitched images were quantified per mouse. Image capturing and quantifications were carried out blinded to genotype.

## Immunohistochemistry and quantification

All paraffin sections used for immunofluorescence staining were deparaffinized with SafeClear (Fisher HealthCare, 23-044192), rehydrated using graded concentrations of ethanol in water (100% EtOH, 95%, 75%, 50%, then dH$_2$O), then subjected to antigen retrieval with a pressure cooker for 5 min in sodium citrate buffer (10 mM sodium citrate, 0.05% Tween 20, pH 6.0). Specimens were washed with a TBS+0.025% Triton X-100 buffer, blocked with 10% normal goat serum at RT for 1 hr, and incubated at 4°C overnight with the appropriate primary antibody. Muscle cells expressing SLN or oxidative fibers were labeled with a rabbit polyclonal anti-SLN antibody (MilliporeSigma, ABT13) or mouse monoclonal Ab to SDHB (Abcam, ab14714), respectively. Primary antibodies were used at a concentration of 1:200 in TBS buffer (with 1% BSA), followed by the appropriate secondary antibody at 1:1000 (diluted in TBS+1% BSA) and incubated for 1 hr at RT. Secondary antibodies used were goat anti-mouse (Invitrogen, Alexa Fluor 594, A21135) and goat anti-rabbit (Invitrogen, Alexa Fluor 488, A11008). All specimens were additionally stained with WGA (wheat germ agglutinin, Alexa Fluor 647, W32466) at a concentration of 1:250 in TBS and mounted with a coverslip using ProLong Gold antifade reagent with DAPI (Invitrogen, P36935). Images were captured with a Keyence BZ-X700 All-in-One fluorescence microscope (Keyence Corp, Itasca, IL, USA). Muscle cell CSA analysis was completed on specimens stained with WGA. At least 1000 cells of the gastrocnemius muscle were quantified per mouse using ImageJ (*Schneider et al., 2012*).

## Western blots analysis

Protein was isolated from skeletal muscle samples using RIPA buffer as previously described (*Rodriguez et al., 2016*). Protein lysates used for SLN immunoblots were boiled for 5 min in a loading buffer (50 mM Tris, 2% SDS, 1% β-ME, 6% glycerol, 0.01% bromophenol blue), while those used

for mitochondrial oxidative phosphorylation complex immunoblots were heated for 3 min at 50°C in the same buffer. Total protein was quantified by BCA assay (Thermo Scientific, 23225), loaded in equal amounts and volume, and run on a 4–20% gradient gel (Bio-Rad, 4561096). Protein was transferred to nitrocellulose or PVDF membrane (SLN and OXPHOS blots, respectively) and blocked in PBS containing 0.2% Tween 20 and 5% non-fat milk for 1 hr, then incubated overnight at 4°C on a shaker with the antibody. SLN was detected using the MilliporeSigma antibody (ABT13, rabbit polyclonal) at a concentration of 1:500. Mitochondrial oxidative phosphorylation complexes were detected with the Abcam OXPHOS cocktail antibody (ab110413, mouse monoclonal) at a concentration of 1:5000. GAPDH was detected using the Proteintech antibody (60004-1-Ig, mouse monoclonal) at a concentration of 1:20,000. Anti-rabbit or anti-mouse secondary antibodies conjugated to HRP were used to recognize the primary antibody. Immunoblots were developed using HRP substrate ECL (GE Healthcare), visualized with a MultiImage III FluorChem Q (Alpha Innotech), and quantified with ImageJ. For the OXPHOS membrane, the blot was re-probed with the GAPDH antibody after stripping with ReBlot Plus Strong Antibody Stripping Solution (MilliporeSigma, 2504).

## Electron microscopy and quantification

Mouse pancreas was dissected and sectioned into six pieces (~1 mm³) for fixation with freshly prepared electron microscopy-grade 2% paraformaldehyde, 2% glutaraldehyde in 100 mM Sorenson's phosphate buffer containing 3 mM $MgCl_2$, pH 7.4, and 1144 mOsm overnight at 4°C on slow rocker. Samples were rinsed with the same buffer containing 3% sucrose, 316 mOsm, and osmicated in 2% osmium tetroxide reduced in 1.5% potassium ferrocyanide for 2 hr at 4°C. Tissue was then rinsed in 100 mM maleate buffer containing 3% sucrose pH 6.2 and en bloc stained in 2% uranyl acetate in maleate buffer for 1 hr at 4°C in the dark. Samples were dehydrated in a graded ethanol series, brought to RT in 70% ethanol, and completely dehydrated in 100% ethanol. Samples were resin-embedded (Epon 812, T. Pella) after a propylene oxide transition step and further infiltrated and cured the next day. Eighty-nm-thin compression free sections were obtained with a Diatome diamond knife (35 degrees). Sections were picked up on 1×2 mm formvar-coated copper slot grids (Polysciences) and further stained with uranyl acetate followed by lead citrate. Grids were examined on a Hitachi H-7600 transmission electron microscope operating at 80 kV. Images of β-cells and acinar cells were digitally captured at magnifications of ×5000 and ×10,000 from six unique locations in the pancreas of each mouse with an AMT XR 50-5 megapixel CCD camera. Within each of the six locations, at least 600 zymogen granules and 200 insulin granules (dense insulin core and vesicle) were measured for CSA quantification using ImageJ software. Analyses were performed on a total of six randomly selected male chow-fed mice (WT, n=3; KO, n=3) and six randomly selected HFD-fed mice (WT, n=3; KO, n=3). Of the six unique locations from a single mouse, values were averaged to generate a mouse average. Therefore, each data point represents a mouse average comprised of at least 3600 zymogen granules, 1200 dense insulin cores, or 1200 insulin vesicles.

## Quantitative real-time PCR analysis

Total RNA was isolated from tissues using Trizol reagent (Thermo Fisher Scientific) according to the manufacturer's instructions. Purified total RNA was reverse-transcribed using an iScript cDNA Synthesis Kit (Bio-Rad). Real-time quantitative PCR (qPCR) analysis was performed on a CFX Connect Real-Time System (Bio-Rad) using iTaq Universal SYBR Green Supermix (Bio-Rad) per manufacturer's instructions. Data were normalized to *36B4* gene (encoding the acidic ribosomal phosphoprotein P0) and expressed as relative mRNA levels using the ΔΔCt method (*Schmittgen and Livak, 2008*). Fold change data were log transformed to ensure normal distribution and statistics were performed. Real-time qPCR primers used are listed in *Supplementary file 15*.

## RNA-sequencing analysis

A total of 50 samples were sequenced with paired-end 50 bp reads (2×50 bp) across two batches that were balanced for genotype (n=40 across liver, brown fat, subcutaneous fat and visceral fat and n=10 of skeletal muscle). Raw sequencing reads from all samples were aligned to a custom concatenated Gencode hg38+mm10 reference genome using HISAT2 2.0.4 (*Kim et al., 2019*). Genes were quantified with featureCounts v1.5.0-p3 (*Liao et al., 2014*) to a custom concatenated hg38+mm10 gtf file within each sample. We retained 24,016 genes with mean RPKM >0.1 across the mouse genome

(N=23,801 genes) and chr21 in the human genome (N=215 genes) above this cut-off, and subsequently dropped two samples with outlier RNA-sequencing quality metrics (one SubQ_MAC21 sample with low gene assignment and high mitochondrial rates and one Skeletal_MAC21 sample with low read alignment rate), leaving a total of 48 samples. High-throughput sequencing data from this study have been submitted to the NCBI Sequence Read Archive (SRA) under accession number PRJNA877694.

We interrogated the effects of MAC21 genotype, tissue, and the statistical interaction between genotype and tissue on the transcriptome, further adjusting for confounders of gene assignment rate and mitochondrial mapping rate, using the limma voom approach (*Law et al., 2014*). Given that four of the tissues were derived from the same animals, we used linear mixed effect using the animal ID as a random intercept. We accounted for multiple testing by controlling for the Benjamini-Hochberg false discovery rate across all mouse genes (since human genes were highly enriched for being DEGs given these defined the genotype effect). As a secondary analysis, we modeled the effect of MAC21 genotype within each tissue (also adjusting for the same two sequencing-derived confounders as the full model) using linear regression (since there were no repeated measures within a tissue). Gene set enrichment analyses were performed using clusterProfiler (*Yu et al., 2012*) and accounted for the false discovery rate.

## Respirometry of frozen tissue samples

Respirometry was conducted on frozen tissue samples to assay for mitochondrial activity as described previously (*Acin-Perez et al., 2020*). Briefly, all tissues were dissected, snap-frozen in liquid nitrogen, and stored at –80°C freezer for later analysis. Samples were thawed in MAS buffer (70 mM sucrose, 220 mM mannitol, 5 mM $KH_2PO_4$, 5 mM $MgCl_2$, 1 mM EGTA, 2 mM HEPES pH 7.4), finely minced with scissors, then homogenized with a glass Dounce homogenizer. The resulting homogenate was spun at 1000 × *g* for 10 min at 4°C. The supernatant was collected and immediately used for protein quantification by BCA assay (Thermo Scientific, 23225). Each well of the Seahorse microplate was loaded with 8 µg of homogenate protein for all tissue types, except for BAT and heart of which 4 µg was loaded. Each biological replicate is comprised of three technical replicates. Samples from all tissues were treated separately with NADH (1 mM) as a complex I substrate or Succinate (a complex II substrate, 5 mM) in the presence of rotenone (a complex I inhibitor, 2 µM), then with the inhibitors rotenone (2 µM) and antimycin A (4 µM), followed by TMPD (0.45 mM) and ascorbate (1 mM) to activate complex IV, and finally treated with azide (40 mM) to assess non-mitochondrial respiration.

## Statistical analyses

All results are expressed as mean ± standard error of the mean (SEM). Statistical analysis was performed with Prism 9 software (GraphPad Software, San Diego, CA, USA). Data were analyzed with two-tailed Student's t-tests or by repeated measures ANOVA. For two-way ANOVA, we performed Sidak's post hoc tests. $p < 0.05$ was considered statistically significant.

## Acknowledgements

The work was funded, in part, by grants from the National Institutes of Health (DK084171 to GWW, HD098540 and HD038384 to RHR), Japan Society for the Promotion of Science (25221308 to MO), and the Core Research for Evolutional Science and Technology (JPMJCR18S4 to YK). The fecal bomb calorimetry analysis was performed at the University of Michigan Animal Phenotyping Core, supported by center grants 1U2CDK135066-01 (Mi-MPMOD) and DK020572 (MDRC). The presented information and its interpretation do not necessarily reflect those of the funding agencies. We thank Nanami Senoo from the Claypool lab for technical advice on conducting the mitochondrial respiration experiments using Seahorse machine.

# Additional information

### Competing interests

Mitsuo Oshimura: M.O. is CEO, employee, and shareholder of Trans Chromosomics, Inc which manages commercial use of the TcMAC21 mouse. We declare that none of the authors has a conflict of interest. The other authors declare that no competing interests exist.

### Funding

| Funder | Grant reference number | Author |
|---|---|---|
| National Institute of Diabetes and Digestive and Kidney Diseases | DK084171 | G William Wong |
| Eunice Kennedy Shriver National Institute of Child Health and Human Development | HD098540 | Roger H Reeves |
| Eunice Kennedy Shriver National Institute of Child Health and Human Development | HD038384 | Roger H Reeves |
| Japan Society for the Promotion of Science | 25221308 | Mitsuo Oshimura |
| Core Research for Evolutional Science and Technology | JPMJCR18S4 | Yasuhiro Kazuki |

The funders had no role in study design, data collection and interpretation, or the decision to submit the work for publication.

### Author contributions

Dylan C Sarver, Conceptualization, Data curation, Formal analysis, Investigation, Visualization, Writing – original draft; Cheng Xu, Investigation; Susana Rodriguez, Feng J Gao, Investigation, Writing - review and editing; Susan Aja, Formal analysis, Investigation, Methodology, Writing - review and editing; Andrew E Jaffe, Data curation, Formal analysis, Methodology; Michael Delannoy, Methodology; Muthu Periasamy, Yasuhiro Kazuki, Mitsuo Oshimura, Resources; Roger H Reeves, Conceptualization, Resources, Funding acquisition, Writing - review and editing; G William Wong, Conceptualization, Formal analysis, Supervision, Funding acquisition, Investigation, Writing – original draft, Project administration

### Author ORCIDs

Yasuhiro Kazuki http://orcid.org/0000-0003-4818-4710
Roger H Reeves http://orcid.org/0000-0002-3581-0850
G William Wong http://orcid.org/0000-0002-5286-6506

### Ethics

All mouse protocols were approved by the Institutional Animal Care and Use Committee of the Johns Hopkins University School of Medicine. All animal experiments were conducted in accordance with the National Institute of Health guidelines and followed the standards established by the Animal Welfare Acts. (animal protocol # MO19M481).

### Decision letter and Author response

Decision letter https://doi.org/10.7554/eLife.86023.sa1
Author response https://doi.org/10.7554/eLife.86023.sa2

## Additional files

### Supplementary files

• Supplementary file 1. Tissue weights of chow-fed euploid (n=7) and MAC21 (n=6) male mice at termination of study. Eu, euploid; BW, body weight; gWAT, gonadal white adipose tissue; iWAT, inguinal white adipose tissue.

• Supplementary file 2. Indirect calorimetry analysis of male (16.5 weeks of age) euploid (n=8) and

MAC21 (n=9) mice fed a standard chow. Eu, euploid; $VO_2$, rate of oxygen consumption; $VCO_2$, rate of carbon dioxide production; RER, respiratory exchange ratio; EE, energy expenditure.

• Supplementary file 3. Tissue weight of high-fat diet-fed male mice (33 weeks of age; fed diet for 16.5 weeks) at termination of the study. Eu, euploid; gWAT, gonadal white adipose tissue; iWAT, inguinal white adipose tissue; BAT, brown adipose tissue.

• Supplementary file 4. Complete blood count of euploid (n=5) and MAC21 (n=6) male mice fed a high-fat diet (50 weeks of age; on diet for 13 weeks). RBC, red blood cell count; HGB, hemoglobin; HCT, hematocrit; MCV, mean corpuscular (erythrocyte) volume; MCH, mean corpuscular hemoglobin; MCHC, mean corpuscular hemoglobin concentration; RET, reticulocyte count; PLT, platelet count; WBC, white blood cells count; NEUT, neutrophil count; LYMPH, lymphocyte count; MONO, monocyte count; EO, eosinophil count; BASO, basophil count; Eu, euploid.

• Supplementary file 5. Indirect calorimetry analysis of male (25 weeks of age) euploid (n=8) and TcMAC21 (n=9) mice fed a high-fat diet (8.5 weeks on diet). Eu, euploid; VOC, rate of oxygen consumption; $VCO_2$, rate of carbon dioxide production; RER, respiratory exchange ratio; EE, energy expenditure.

• Supplementary file 6. List of all differentially expressed genes (DEGs) in the gonadal white adipose tissue (gWAT) of euploid and MAC21 male mice fed a high-fat diet housed at 22°C. Lists include: detected human genes and all significantly up- and downregulated mouse genes.

• Supplementary file 7. List of all differentially expressed genes (DEGs) in the inguinal white adipose tissue (iWAT) of euploid and MAC21 male mice fed a high-fat diet housed at 22°C. Lists include: detected human genes and all significantly up- and downregulated mouse genes.

• Supplementary file 8. List of all differentially expressed genes (DEGs) in the liver of euploid and MAC21 male mice fed a high-fat diet housed at 22°C. Lists include: detected human genes and all significantly up- and downregulated mouse genes.

• Supplementary file 9. List of all differentially expressed genes (DEGs) in the brown adipose tissue (BAT) of euploid and MAC21 male mice fed a high-fat diet housed at 22°C. Lists include: detected human genes and all significantly up- and downregulated mouse genes.

• Supplementary file 10. List of all differentially expressed genes (DEGs) in the skeletal muscle (gastrocnemius) of euploid and MAC21 male mice fed a high-fat diet housed at 22°C. Lists include: detected human genes and all significantly up- and downregulated mouse genes.

• Supplementary file 11. RNA-sequence data from skeletal muscle (gastrocnemius) showing expression values for atrophy-related genes, contractile and structural genes, fibrosis and injury repair-related genes, and muscle wasting-related genes.

• Supplementary file 12. Indirect calorimetry analysis in thermoneutral conditions (31°C) after 2-week acclimatization period of male (55 weeks of age) euploid (n=5) and MAC21 (n=4) mice fed a high-fat diet (18 weeks on diet). Eu, euploid; $VO_2$, rate of oxygen consumption; $VCO_2$, rate of carbon dioxide production; RER, respiratory exchange ratio; EE, energy expenditure.

• Supplementary file 13. Tissue weights of male MAC21 (n=4) and euploid (n=5) mice (55 weeks of age) housed at thermoneutrality (31°C) and fed a high-fat diet (HFD, 18 weeks on diet). Eu, euploid; gWAT, gonadal white adipose tissue; iWAT, inguinal white adipose tissue; BAT, brown adipose tissue.

• Supplementary file 14. List of all differentially expressed genes (DEGs) in the skeletal muscle (gastrocnemius) of euploid and MAC21 male mice fed a high-fat diet housed at thermoneutrality (31°C). Lists include: detected human genes and all significantly up- and downregulated mouse genes.

• Supplementary file 15. List of primers used for all qPCR analysis of gene expression.

• MDAR checklist

### Data availability
High-throughput sequencing data from this study have been submitted to the NCBI Sequence Read Archive (SRA) under accession number PRJNA877694.

The following dataset was generated:

| Author(s) | Year | Dataset title | Dataset URL | Database and Identifier |
|---|---|---|---|---|
| Sarver DC, Xu C, Rodriguez S, Aja S, Jaffe AE, Gao FJ, Delannoy M, Periasamy M, Kazuki Y, Oshimura M, Reeves RH, Wong GW | 2023 | Gene expression analyses of TcMAC21 mice | http://www.ncbi.nlm.nih.gov/bioproject/?term=PRJNA877694 | NCBI BioProject, PRJNA877694 |

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
