## [Editor Report]

This important paper provides new insight into the effect of extra-copies of a chromosome, thus aneuploidy, on body metabolisms in mammals. The authors used various solid analyses on the metabolisms and physiology of the transgenic mouse with most of human chromosome 21 and presented convincing results to support the authors' claims. The work would be of interest to researchers who work on the physiology and biochemistry of body metabolisms in mammals.

---

## [Decision Letter]

**Decision letter after peer review:**

Thank you for submitting your article "Hypermetabolism in mice carrying a near complete human chromosome 21" for consideration by *eLife*. Your article has been reviewed by 3 peer reviewers, including Akira Shinohara as the Reviewing Editor and Reviewer #1, and the evaluation has been overseen byCarlos Isales as the Senior Editor.

*Reviewer #1 (Recommendations for the authors):*

This paper nicely characterized the physiology and metabolisms of a transgenic mouse with most of human chromosome 21 with 199 protein-coding genes (PCGs) among 213 on the whole chromosome, called TcMAC21 under the normal condition as well as a condition with high-fat diet (HFD). By doing various physiological and metabolic analyses of TcMAC21 mice including transcriptome analysis of both human and mouse genes in the TcMAC21, the authors showed that TcMAC21 mice are lean, hyperactive, and hypermetabolic, which is caused by an increased protein level of sarcolipin, a membrane protein regulating the activity of sarcoplasmic reticulum Ca2+ ATPase (SERCA) in skeleton muscles, thus futile SERC activity. All experiments were carried out well with control and the results presented in the paper are very much convincing.

One of the weaknesses of the paper is that the authors did not provide any molecular insight on how the presence of human chromosome 21 in skeleton muscles increases the sarcolipin. However, this would be a target of future study.

*Reviewer #2 (Recommendations for the authors):*

Fiber types need to be shown across the genotypes for the muscles examined.

Declare what is causing hypermetabolism: resting muscle or active muscle; or a combination of both? Or declare you don't know the proportions.

Many corrections to the description of Ca handling are required, see above.

Figure 5C: Correct spelling: "Ca Handling".

[Editors' note: further revisions were suggested prior to acceptance, as described below.]

Thank you for resubmitting your work entitled "Hypermetabolism in mice carrying a near complete human chromosome 21" for further consideration by *eLife*. Your revised article has been evaluated by Carlos Isales (Senior Editor) and a Reviewing Editor.

The manuscript has been improved but there are some remaining issues that need to be addressed, as outlined below:

Based on the #3 reviewer's comments, it would be nice to soften the conclusion by adding some sentences and revising the text.

*Reviewer #1 (Recommendations for the authors):*

The authors clearly addressed my concerns as well as ones by the #2 reviewer.

*Reviewer #2 (Recommendations for the authors):*

I am happy with the responses to my points.

*Reviewer #3 (Recommendations for the authors):*

Sarver et al. revisions have improved the paper, however, the data and their responses are not satisfactory. Their interpretation of the data is too strong and the tone needs to be softened. Below are my remaining criticisms.

Dividing EE by lean mass

It is incorrect to normalize EE to lean mass if this parameter is different between groups (PMID: 20103710; PMID: 22205519). As noted clearly in PMID: 22205519, the problem of dividing by lean mass is that the division by lean mass overcompensates for the mass effect. The key conclusion of this paper is that ANCOVA or generalized linear modeling (PMID: 22205519 and references 37-39,43 within) is the most appropriate statistical approach to accommodate discrete (genotype) and continuous (body mass) traits, rather than using a simple division by BW or lean BW. The data where the authors show EE divided by lean mass is completely misleading. Only the ANCOVA should be shown.

Regarding EE by ANCOVA

The authors state in their response: "we have now also included the ANCOVA data (Figure 2D-F and Figure 4D-F) where we used body weight as a covariate as recommended (PMID: 22205519). The results clearly indicate that the TcMAC21 mice have significantly higher EE compared to the euploid controls." I do not understand how the authors can say these results are clear. The ANCOVA of EE (FIGURE 2D, FIGURE 2F and FIGURE 7F) look like they lie on a common line in relation to lean mass. The ANCOVA in FIGURE 2E looks more promising, but it tracks very well with physical activity (i.e. the activity levels are strikingly higher in 2E compared to 2D and 2F). Regarding the ANCOVAs, I cannot find any easily seen statistics that were undertaken to indicate that these effects that are shown indicate an increase in EE that is independent of body composition changes. I do not see any statistics reported for the ANCOVA in the methods or in the figure legends. The word "ANCOVA" appears 4 times in the paper: (1) Figure 2 legend; (2) Figure 4 legend; (3) Fig7 legend; (4) Methods (indirect calorimetry section). So, how can the authors assert that "we have now also included the ANCOVA data (Figure 2D-F and Figure 4D-F) where we used body weight as a covariate as recommended (PMID: 22205519). The results clearly indicate that the TcMAC21 mice have significantly higher EE compared to the euploid controls"?

Regarding mechanistic claims about SERCA-based calcium cycling

The tone about SERCA-based calcium cycling being causal for any metabolic phenotype needs to be toned down substantially. As, the authors note, they cannot determine if their proposed hypermetabolism is arising from movement or SERCA, but considering these associative data and the complete lack of direct analysis of SERCA activity in this paper, it is imperative that the authors tone down their claims about futile calcium cycling. For example, they make strong claims such as: (1) "Unexpectedly, we discovered that TcMAC21 mice have all the hallmarks of hypermetabolism, driven by elevated mitochondrial respiration and futile sarco(endo)plasmic reticulum ca^2+^ ATPase (SERCA) pump activity in the skeletal muscle." (2) "All together, these data suggest that SLN-mediated futile SERCA activity in skeletal muscle is likely the dominant driver underlying the hypermetabolism phenotypes seen in TcMAC21 mice". (3) "Having established that SLN-mediated futile SERCA activity underlies hypermetabolism in TcMAC21 mice….". These statements are quite strong. However, there is no data that directly measures futile SERCA activity and there is no data that link SERCA-mediated calcium cycling to the metabolic effects of TcMAC21 mice. Other than SLN expression, there is no evidence that any futile thermogenic cycle is involved. The authors mention in the responses that "ATP-consuming futile activity of the SERCA pump is presumably happening in resting muscle" but they provide no causal evidence other than mRNA and protein expression. In contrast, there is clear evidence that TcMAC21 mice are more physically active and that the changes in ANCOVA-based EE track well with movement (see above). There is also indications that there is a higher degree of respiratory chain levels, which could be secondary to elevated physical activity – akin to exercise training. This is in contrast to the differences in EE between TcMAC21 mice and controls, where the effect size is minimized at thermoneutrality, even though a major tenet is that SLN expression is high in a temp-independent manner.

Why not anesthetize mice and measure EE to determine if physical activity is driving the increased metabolic rate?

Food intake measurements are not clear

Cumulative food intake would be more informative. The food intake data is plotted per day. How was this measured? For example, was some random day chosen to quantify food consumption, or was the food consumption quantified over the duration of the study and then normalized by the duration to get a daily average? If the latter, then the authors are correct and this could support their argument that food intake is not contributing to the phenotype. If the former, this is questionable of whether the particular day they chose to measure was representative of the entire duration of the intervention.

EE does not seem to be higher at TN

The authors note that "TcMAC21 mice still have significantly higher EE compared to euploid controls when housed at thermoneutrality." First, there are no statistics that I can find in the paper that support the authors' claim. Second, the ANCOVA in FIGURE 7F looks like the regression lines of the EE between the two groups lie on a common line in relation to lean mass. Thus, the authors' claim "the TcMAC21 mice never reduce their EE to the level of euploid controls when housed at thermoneutrality" does not appear correct. Moreover, EE should not be plotted as a histogram, which is what the authors have done in FIGURE 7F to conduct their stats. The statistics should be a 2-way ANOVA for 7F (although beware as the data is improperly normalized by lean mass).

GTT

In my first review, I mentioned that GTT should be normalized to lean mass, since that is very different between TcMAC21 mice and controls. Guidelines have been written about this (PMID: 29143855). However, the authors responded by saying that: "The dose of glucose injection in GTT based on mouse weight is widely and extensively practiced across the metabolic community." I am bewildered by the fact that the authors think that dividing EE by lean mass is acceptable, but then don't think normalizing for lean mass for GTT is appropriate. The response to the ITT being better in TcMAC21, despite getting less of a dose, is a decent argument that the authors provide. Unfortunately, the basal blood glucose levels of the ITT in FIGURE 3K were different to begin with. The authors could normalized the baseline to 100%. This will decrease the difference, but maybe statistical significance can still be reached. Or you could use AUC to correct for baseline differences.

The authors say they "sought to uncover the physiological mechanisms responsible for TcMAC21 resistance to weight gain…". The authors mention that their data strongly support hypermetabolism being the cause of the lean phenotype seen in TcMAC21 mice. However, this has not really been shown. The massive reductions in lean mass have erroneously made it seem like EE is elevated, and the ANCOVAs do not look very promising by eye, but importantly there is no statistical test done on the ANCOVA that I could find. The p value should be placed on the graph to make it clear. In any case, if the authors want to uncover the physiological mechanisms responsible for TcMAC21 resistance to weight gain, EE needs to be measured PRIOR to a change in body composition. If hypermetabolism (i.e. physical activity) is causal, then it should precede the changes in body weight/composition that occur later in life. The authors note in their paper that TcMAC21 mice are born at the same weight as their euploid littermates, but by 3 months of age weigh significantly less. Is there a time-window where TcMAC21 mice can be compared to controls, when their weights are identical? If EE is higher at that early time, this would be strong evidence that hypermetabolism drives the phenotypes associated after weights have diverged.

---

## [Author Response]

Reviewer #2 (Recommendations for the authors):Fiber types need to be shown across the genotypes for the muscles examined.

In the qPCR data as shown in Figure 6C, we have profiled many genes associated with slow- and fast-twitched muscle fibers in gastrocnemius, and little if any changes were noted. At least at the level of the transcript, there is no indication of fiber type switching in gastrocnemius muscle. However, we did not perform the same qPCR analyses for all the other muscle types we isolated (EDL, quadriceps, plantaris, soleus, and tongue). The main reason for this is that we have used all of these muscle in our respirometry analysis shown in Figure 6O-Q and Figure 6—figure supplement 4-9. We did not have any leftover muscle tissue to profile muscle fiber types.

Declare what is causing hypermetabolism: resting muscle or active muscle; or a combination of both? Or declare you don't know the proportions.

In short, we do not know the proportions. The TcMAC21 mice have persistent uncoupling of the SERCA pumps due to chronic overexpression of SLN. However, the mice also are hyperactive in both light and dark cycle, though the hyperactivity is much more pronounced in the dark cycle. Since both SERCA pump uncoupling and hyperactivity are both present at all times of the day, it is difficult to disentangle the relative contribution of each to the hypermetabolic phenotype.

Many corrections to the description of Ca handling are required, see above.

We have fixed all the issues raised by the reviewer.

Figure 5C: Correct spelling: "Ca Handling".

We have fixed this typo in Figure 6C.

[Editors' note: further revisions were suggested prior to acceptance, as described below.]

Reviewer #3 (Recommendations for the authors):Sarver et al. revisions have improved the paper, however, the data and their responses are not satisfactory. Their interpretation of the data is too strong and the tone needs to be softened. Below are my remaining criticisms.

We have made revisions to the manuscript text accordingly. We have softened the interpretation of our data as recommended.

Dividing EE by lean massIt is incorrect to normalize EE to lean mass if this parameter is different between groups (PMID: 20103710; PMID: 22205519). As noted clearly in PMID: 22205519, the problem of dividing by lean mass is that the division by lean mass overcompensates for the mass effect. The key conclusion of this paper is that ANCOVA or generalized linear modeling (PMID: 22205519 and references 37-39,43 within) is the most appropriate statistical approach to accommodate discrete (genotype) and continuous (body mass) traits, rather than using a simple division by BW or lean BW. The data where the authors show EE divided by lean mass is completely misleading. Only the ANCOVA should be shown.

We agree with the reviewer that the ANCOVA analysis is very helpful in understanding the metabolic phenotypes of mice. We have provided ANCOVA analyses, including the statistics, in the revision. In our study, EE normalized to lean mass or presented by ANCOVA gives the same results, namely that the TcMAC21 mice have significantly higher energy expenditure. For the sake of transparency and thoroughness, we provided both types of data for the reader to see, as both types of analyses yielded the same results. The 24 hr tracing allows readers to appreciate that TcMAC21 mice have higher EE across the entire circadian cycle (light and dark), as well as across three different metabolic states (ad libitum fed, fasting, refeeding). Even the reviewer had specifically requested that we show the 24 tracing for the thermoneutral data (Figure 7) in the previous round of review. For this reason, we would like to keep all the figure panels. However, to acknowledge the important point raised by the reviewer, we have now clearly stated in the result section (in the revision) that EE normalized to lean mass can result in an overestimation and that ANCOVA results are included to further support our observations that TcMAC21 mice have significantly higher energy expenditure.

Regarding EE by ANCOVAThe authors state in their response: "we have now also included the ANCOVA data (Figure 2D-F and Figure 4D-F) where we used body weight as a covariate as recommended (PMID: 22205519). The results clearly indicate that the TcMAC21 mice have significantly higher EE compared to the euploid controls." I do not understand how the authors can say these results are clear. The ANCOVA of EE (FIGURE 2D, FIGURE 2F and FIGURE 7F) look like they lie on a common line in relation to lean mass. The ANCOVA in FIGURE 2E looks more promising, but it tracks very well with physical activity (i.e. the activity levels are strikingly higher in 2E compared to 2D and 2F). Regarding the ANCOVAs, I cannot find any easily seen statistics that were undertaken to indicate that these effects that are shown indicate an increase in EE that is independent of body composition changes. I do not see any statistics reported for the ANCOVA in the methods or in the figure legends. The word "ANCOVA" appears 4 times in the paper: (1) Figure 2 legend; (2) Figure 4 legend; (3) Fig7 legend; (4) Methods (indirect calorimetry section). So, how can the authors assert that "we have now also included the ANCOVA data (Figure 2D-F and Figure 4D-F) where we used body weight as a covariate as recommended (PMID: 22205519). The results clearly indicate that the TcMAC21 mice have significantly higher EE compared to the euploid controls"?

We apologize that the ANCOVA statistics were not included in the previous revision. We have now included the ANCOVA statistic in Figure 2D, Figure 4D, and Figure 7F, calculated based on the method outlined in the article (PMID: 22205519, Supplementary Note 4). Figure 2D and 2F show diverging regression lines, indicating that as body weight increases there will be no point at which the EE of TcMAC21 mice is not higher than their euploid controls. As exemplified in Figure 1 and 2 of the article (PMID: 22205519), the TcMAC21 mice fit the category where the animals have higher energy expenditure compared to control mice. Even a modest increase in EE can result in significant weight loss over time. Because EE is continuously higher in TcMAC21 relative to euploid controls (regardless of the circadian cycle and metabolic states), this persistently higher EE contributes to the lean phenotype we observed.

Regarding mechanistic claims about SERCA-based calcium cyclingThe tone about SERCA-based calcium cycling being causal for any metabolic phenotype needs to be toned down substantially. As, the authors note, they cannot determine if their proposed hypermetabolism is arising from movement or SERCA, but considering these associative data and the complete lack of direct analysis of SERCA activity in this paper, it is imperative that the authors tone down their claims about futile calcium cycling. For example, they make strong claims such as: (1) "Unexpectedly, we discovered that TcMAC21 mice have all the hallmarks of hypermetabolism, driven by elevated mitochondrial respiration and futile sarco(endo)plasmic reticulum ca^2+^ ATPase (SERCA) pump activity in the skeletal muscle." (2) "All together, these data suggest that SLN-mediated futile SERCA activity in skeletal muscle is likely the dominant driver underlying the hypermetabolism phenotypes seen in TcMAC21 mice". (3) "Having established that SLN-mediated futile SERCA activity underlies hypermetabolism in TcMAC21 mice….". These statements are quite strong. However, there is no data that directly measures futile SERCA activity and there is no data that link SERCA-mediated calcium cycling to the metabolic effects of TcMAC21 mice. Other than SLN expression, there is no evidence that any futile thermogenic cycle is involved. The authors mention in the responses that "ATP-consuming futile activity of the SERCA pump is presumably happening in resting muscle" but they provide no causal evidence other than mRNA and protein expression. In contrast, there is clear evidence that TcMAC21 mice are more physically active and that the changes in ANCOVA-based EE track well with movement (see above). There is also indications that there is a higher degree of respiratory chain levels, which could be secondary to elevated physical activity – akin to exercise training. This is in contrast to the differences in EE between TcMAC21 mice and controls, where the effect size is minimized at thermoneutrality, even though a major tenet is that SLN expression is high in a temp-independent manner.Why not anesthetize mice and measure EE to determine if physical activity is driving the increased metabolic rate?

Per reviewer request, we have toned down the interpretation of our data in the revised manuscript. We agree that we did not provide direct evidence that SLN is the sole contributing factor to the hypermetabolic phenotype observed in the TcMAC21 mice. In our previous revision (based on the comments by reviewer 2), we did indicate that we do not know the relative contribution of elevated metabolism vs. increased physical activity to the overall lean phenotype of TcMAC21 mice. However, some of our evidence suggest that SLN-mediated futile cycling, at least in part, drives the hypermetabolism phenotype we observed; for example, in Figure 6O-Q, we show that mitochondrial respiration is significantly elevated in several different muscle fiber types, including quadriceps, EDL, gastrocnemius, and plantaris. Because SLN overexpression will result in increased ATP utilization (based on published studies referenced in our manuscript), this presumably necessitates an increase in mitochondrial respiration in the skeletal muscle to meet the demand created by the SLN-mediated futile SERCA pump activity. We do not know whether this reflects skeletal muscle adaptation to increased physical activity. As a future follow up experiment, we agree that anesthetizing the mice and measuring EE would be helpful in teasing out the contribution of SLN-overexpression on the hypermetabolic phenotype of TcMAC21 mice. However, the suggested experiment can be technically challenging to perform because the mice need to be housed in the metabolic cage (CLAMS) and at the same time be subjected to prolonged anesthesia (which is known to depress respiration, heart rate, and metabolism).

Food intake measurements are not clearCumulative food intake would be more informative. The food intake data is plotted per day. How was this measured? For example, was some random day chosen to quantify food consumption, or was the food consumption quantified over the duration of the study and then normalized by the duration to get a daily average? If the latter, then the authors are correct and this could support their argument that food intake is not contributing to the phenotype. If the former, this is questionable of whether the particular day they chose to measure was representative of the entire duration of the intervention.

Food intake was measured while the mice were in the CLAMS (comprehensive laboratory animal monitoring system). Food intake was measured alongside VO_2_, VCO_2_, RER, EE, and activity. The general process can be found in the Methods section under indirect calorimetry. Briefly, data were collected for three days to confirm mice were acclimatized to the calorimetry chambers (indicated by stable body weights, food intakes, and diurnal metabolic patterns), and data were analyzed from the subsequent three days. Data from mice were collected with ad libitum access to food, throughout the fasting process, and in response to refeeding. It is more common for mice to under eat than over eat while in the CLAMS chambers. Therefore, it is less likely that the food intake assessment grossly overestimated the food intake levels of TcMAC21 mice. For the sake of transparency, we provided the food intake data we collected for 6 consecutive days in the CLAMS. In the chow-fed group, the TcMAC21 mice tended to eat a bit more than the euploid controls. For the high-fat group, except day 1 where TcMAC21 eat a little less than euploid controls, the rest of the days the TcMA21 mice eat either a similar amount or significantly more (Day 6, refeeding after a 24 fast). Even though TcMAC21 mice weigh substantially less than the euploid controls, they eat as much as the control mice. This data is consistent with the notion that TcMAC21 mice have significantly higher energy expenditure and therefore need to increase caloric intake to fuel the higher energy metabolism.

**Author response table 1. sa2table1:** 

	Food intake (kcal/day)		
Chow-fed	Euploid	TcMAC21	*p-*value
Day 1	12.93±1.75	13.39±1.07	0.52
Day 2	11.95±2.41	13.16±1.47	0.23
Day 3	12.20±2.60	13.32±1.68	0.3
Day 4	12.74±2.56	13.41±2.26	0.57
Day 5 (fasted)	0	0	
Day 6 (Refed)	17.55±2.97	17.32±1.46	0.83
			
	Food intake (kcal/day)		
HFD-fed	Euploid	TcMAC21	*p-*value
Day 1	15.52±3.68	13.00±1.71	0.06
Day 2	12.30±5.6	12.0±2.22	0.8
Day 3	14.67±5.90	13.80±2.60	0.69
Day 4	15.28±3.40	16.15±2.48	0.55
Day 5 (fasted)	0	0	
Day 6 (Refed)	13.04±2.46	17.74±2.30	0.001

EE does not seem to be higher at TNThe authors note that "TcMAC21 mice still have significantly higher EE compared to euploid controls when housed at thermoneutrality." First, there are no statistics that I can find in the paper that support the authors' claim. Second, the ANCOVA in FIGURE 7F looks like the regression lines of the EE between the two groups lie on a common line in relation to lean mass. Thus, the authors' claim "the TcMAC21 mice never reduce their EE to the level of euploid controls when housed at thermoneutrality" does not appear correct. Moreover, EE should not be plotted as a histogram, which is what the authors have done in FIGURE 7F to conduct their stats. The statistics should be a 2-way ANOVA for 7F (although beware as the data is improperly normalized by lean mass).

The statistics for Figure 7F (the bar graphs) were calculated using 2-way ANOVA. We included this information in the figure legend. We have also included the ANCOVA statistics in Figure 7F. The ANCOVA of Figure 7F shows separated and divergent regression lines. Although TcMAC21 mice do still appear to have a higher EE at thermoneutrality relative to euploid controls, because of the small sample size, the *P*-value (0.069) fell short of reaching statistical significance. We have now revised our manuscript text accordingly. Much of the work was carried out during the first two years of the pandemic where we had to navigate many restrictions imposed on mouse studies. We acknowledge that the thermoneutral study was underpowered and this is clearly a limitation. We did the best we could in light of the major challenges of breeding this new mouse line. We can only use the females for breeding as the males are infertile (similar to other Down Syndrome mouse models). As stated in Supplementary note 5 of the article (PMID: 22205519) “…virtually every study of energy balance yet performed, including all by our own groups, have probably been underpowered to detect subtle effects of the treatment on energy balance.”. The concluding paragraph in the last page of the Supplementary note of the article (PMID: 22205519) also stated: “…we note that referees and editors also need to be realistic in reviewing papers not to insist on completely unrealistic power for this type of analysis. Otherwise nothing will be published.”

GTTIn my first review, I mentioned that GTT should be normalized to lean mass, since that is very different between TcMAC21 mice and controls. Guidelines have been written about this (PMID: 29143855). However, the authors responded by saying that: "The dose of glucose injection in GTT based on mouse weight is widely and extensively practiced across the metabolic community." I am bewildered by the fact that the authors think that dividing EE by lean mass is acceptable, but then don't think normalizing for lean mass for GTT is appropriate. The response to the ITT being better in TcMAC21, despite getting less of a dose, is a decent argument that the authors provide. Unfortunately, the basal blood glucose levels of the ITT in FIGURE 3K were different to begin with. The authors could normalized the baseline to 100%. This will decrease the difference, but maybe statistical significance can still be reached. Or you could use AUC to correct for baseline differences.

Since muscle mass scales with body weight, a heavier mouse will have more lean mass. A quick survey of the recent literature (PMID: 29724723, PMID: 33571454, PMID: 22961109, PMID: 36329217, PMID: 28844881, PMID: 35995995, PMID: 35677645, PMID: 30605666, PMID: 34819664, PMID: 29466739, PMID: 33691105, PMID: 31875646) shows a number of studies that have used glucose dose based on body weight and not lean mass even though there are significant differences (in some cases large differences) in body weight, and in some of these studies the lean mass was also significantly different. Although not an exhaustive search, the authors have been unable to find a study where the dose of glucose injected (in GTT) is based on lean mass. The dose of glucose injected (for GTT) based on body weight likely reflects the fact that the insulin-regulated glucose transporter GLUT4 is mainly expressed in skeletal muscle and adipose tissue. Adipose-specific knockout of GLUT4 actually has similar metabolic phenotypes as the skeletal muscle-specific knockout of GLUT4 (PMID: 12788932), thus underscoring the relative importance of insulin-regulated glucose uptake in adipose tissue vs. skeletal muscle. Therefore, it provides a rationale to give a dose of glucose based on body weight (accounting for glucose uptake in both lean and fat mass). It is true that the basal blood glucose does not start at the exact same place as represented in the ITT graph of Figure 3K, but this is also not significantly different between genotype. As requested, we included the AUC graph as Author response image 1. The AUC data is consistent with the ITT tracing, indicating greater insulin sensitivity in TcMAC21 mice relative to euploid controls.

**Author response image 1. sa2fig1:** 

For insulin sensitivity, we have multiple independent lines of evidence in addition to the GTT data: (1) The ITT data that directly test for insulin action. Even though TcMAC21 received a much smaller dose of insulin (dose based on body weight), glucose clearance was significantly greater in TcMAC21 mice (Fig, 3K). (2) Overnight fasting blood glucose and insulin levels were significantly lower in TcMAC21 mice (Figure 3G). (3) HOMA-IR (insulin resistant index) was significantly lower in TcMAC21 mice (Figure 3H). (4) In the refeeding period following an overnight fast, the TcMAC21 mice blood glucose rose to the same level as the euploid control (Figure 3L), but the rise in insulin levels in response to refeeding was substantially lower in TcMAC21 mice. This indicates that a much lower amount of insulin is required for TcMAC21 to handle the rise in blood glucose after refeeding. Altogether, these data provide strong support that TcMAC21 mice are indeed more insulin sensitive compared to euploid controls.

The authors say they "sought to uncover the physiological mechanisms responsible for TcMAC21 resistance to weight gain…". The authors mention that their data strongly support hypermetabolism being the cause of the lean phenotype seen in TcMAC21 mice. However, this has not really been shown. The massive reductions in lean mass have erroneously made it seem like EE is elevated, and the ANCOVAs do not look very promising by eye, but importantly there is no statistical test done on the ANCOVA that I could find. The p value should be placed on the graph to make it clear. In any case, if the authors want to uncover the physiological mechanisms responsible for TcMAC21 resistance to weight gain, EE needs to be measured PRIOR to a change in body composition. If hypermetabolism (i.e. physical activity) is causal, then it should precede the changes in body weight/composition that occur later in life. The authors note in their paper that TcMAC21 mice are born at the same weight as their euploid littermates, but by 3 months of age weigh significantly less. Is there a time-window where TcMAC21 mice can be compared to controls, when their weights are identical? If EE is higher at that early time, this would be strong evidence that hypermetabolism drives the phenotypes associated after weights have diverged.

There was no “massive reductions in lean mass…” in TcMAC21 mice. This would constitute a pathological condition (aging-associated sarcopenia, injury-associated muscle atrophy, etc). The TcMAC21 mice weigh a lot less. Since skeletal muscle mass scales with body weight, the muscle mass of TcMAC21 mice scale appropriately with their lean body weight, as highlighted by the % lean mass data (Figure 2B and 3A). In fact, the relative lean mass (% of body weight) was higher in HFD-fed TcMAC21 mice relative to euploid controls (Figure 3B). Similarly, a person that weighs 300 pounds will have more muscle mass than someone that weighs 150 pounds; in each case, the muscle mass scales accordingly in order to support different amounts of body weight.

We have now included the ANCOVA statistics in Figure 2, 4 and 7. Except for the thermoneutral data that fell short of reaching statistical significance (due to small sample size), all the other ANCOVA data were highly significant.

We agree that finding a time in which the body weights of TcMAC21 and euploid control mice are equal would be hugely beneficial to uncover more about the hypermetabolism of the TcMAC21mice. Unfortunately, while fed a standard chow or a high-fat diet there is no time window beyond P1 that TcMAC21 mice weigh the same as their euploid littermates. Assessing EE of young pups (while they are still suckling) is technically challenging to conduct. The EE data needs to be collected in a specialized metabolic cage where the animal is individually housed and this is not feasible when the pups are still suckling.